# Functional networks of inhibitory neurons orchestrate synchrony in the hippocampus

**Marco Bocchio[1,2]☯, Artem Vorobyev[1]☯, Sadra Sadeh[3], Sophie Brustlein[1], Robin Dard[1], Susanne Reichinnek[1], Valentina Emiliani[4], Agnes Baude[1], Claudia Clopath[5], Rosa Cossart [1]***

**1** Aix Marseille, University, Inserm, INMED, Turing Center for Living Systems, Marseille, France, **2** Department of Psychology, Durham University, Durham, United Kingdom, **3** Department of Brain Sciences, Imperial College London, London, United Kingdom, **4** Wavefront-Engineering Microscopy Group, Photonics Department, Vision Institute, Sorbonne University, INSERM, CNRS, Paris, France, **5** Department of Bioengineering, Imperial College London, London, United Kingdom

☯ These authors contributed equally to this work.
* rosa.cossart@inserm.fr

## Abstract

Inhibitory interneurons are pivotal components of cortical circuits. Beyond providing inhibition, they have been proposed to coordinate the firing of excitatory neurons within cell assemblies. While the roles of specific interneuron subtypes have been extensively studied, their influence on pyramidal cell synchrony in vivo remains elusive. Employing an all-optical approach in mice, we simultaneously recorded hippocampal interneurons and pyramidal cells and probed the network influence of individual interneurons using optogenetics. We demonstrate that CA1 interneurons form a functionally interconnected network that promotes synchrony through disinhibition during awake immobility, while preserving endogenous cell assemblies. Our network model underscores the importance of both cell assemblies and dense, unspecific interneuron connectivity in explaining our experimental findings, suggesting that interneurons may operate not only via division of labor but also through concerted activity.

## Introduction

Inhibitory interneurons are indispensable constituents of cortical circuits, regulating the computational processes within neural networks. Through the release of the inhibitory neurotransmitter GABA onto various subcellular compartments of excitatory neurons, interneurons finely modulate input integration and spike generation, exerting control over both individual neuron activity and population dynamics [1]. Their multifaceted roles include the generation of network oscillations across different frequencies and the facilitation of synchrony among groups of excitatory neurons through mechanisms such as disinhibition and rebound excitation [2–8]. Consequently, interneurons emerge as pivotal regulators, shaping the dynamic interplay and temporal boundaries of co-active principal neurons.

This regulatory influence of interneurons on neuronal synchrony gains particular significance in the CA1 region of the adult hippocampus. The dynamics of CA1 are frequently

**Data Availability Statement:** Spontaneous calcium imaging data (Figs 1 and 4) are available at https://zenodo.org/records/11491806. All-optical (imaging and optogenetics) data (Figs 2 and 4) are available

at https://zenodo.org/records/11500028. Source codes used for analyzing the data and generating the network model are available at https://zenodo.org/records/12899622 and https://gitlab.com/cossartlab/bocchio-vorobyev-et-al-2023/.

**Funding:** RC was supported by the European Research Council (ERC-2014-CoG, grant no. 646925 and ERC-2020-SyG, grant no. 951330), by the Fondation Bettencourt Schueller, the Fondation Roger de Spoelberch, by the French National Research Agency (ANR, grant no. ANR-14-CE13-0016) and Centre national de la recherche scientifique (CNRS). MB was supported by the European Union (H2020-MSCA-IF-2017, grant no. 794861). AV was supported by the European Union (H2020-MSCA-IF-2017, grant no. 765549). SS was supported by the Wellcome Trust (grant no. 225412/Z/22/Z). VE was supported by the Fondation Bettencourt Schueller (Prix Coups d'Elan pour la Recherche Française) and the European Research Council (ERC-2019-AdG; grant no. 885090). The funders had no role in study design, data collection and analysis, decision to publish, or preparation of the manuscript.

**Competing interests:** The authors have declared that no competing interests exist.

**Abbreviations:** AP, anteroposterior; CR, calretinin; DV, dorsoventral; FOV, field of view; ICA, independent component analysis; ML, mediolateral; PBS, phosphate-buffered saline; PCA, principal component analysis; PV, parvalbumin; ROI, region of interest; RT, room temperature; SCE, synchronous calcium event; SWR, sharp-wave ripple; VIP, vasoactive intestinal peptide.

conceptualized within the framework of 'cell assemblies' [9–12], groups of neurons that coordinate their activity and are considered fundamental units supporting memory processes [13–15]. Recent investigations underscore the pivotal role of local circuits within CA1 in modulating both single neuron and network activity in response to external inputs [16–18], suggesting a significant contribution of inhibition to these mechanisms. Unlike cortical regions where recurrent connectivity relies predominantly on glutamatergic excitatory synapses, CA1 interneurons play a substantial role in mediating recurrent connections among principal neurons [19]. Interneurons in CA1 are densely interconnected with pyramidal cells, receiving feedback inputs from a wide array of local pyramidal cells [17,20–27]. Conversely, the majority of local inputs onto CA1 pyramidal cells originate from interneurons [18]. In addition, CA1 interneurons are densely connected with each other. This dense interconnectivity extends beyond subtypes specialized in targeting other interneurons, such as vasoactive intestinal peptide (VIP) or calretinin (CR)-expressing interneurons [28]. This tightly woven network of inhibitory connections may explain the profound impact of stimulating a single "hub neuron" on network activity observed in vitro [29]. However, it remains uncertain whether this prominent network influence of individual CA1 interneurons also holds true in adult, in vivo contexts, or if it is restricted to specific conditions and rare inhibitory subtypes like hub cells.

The prevailing assumption posits that interneurons collaborate through a division of labor to control pyramidal cell inputs, spike timing, and ultimately their recruitment in cell assemblies. Recent studies using optogenetics have causally established that the collective activation or inhibition of similar types of CA1 interneurons including parvalbumin (PV), (CCK) or VIP-containing interneurons, can trigger or prevent the generation of different oscillations and synchronous spiking among pyramidal cells [30–35]. However, this categorization of interneurons into subtypes does not fully explain how their activity is coordinated at the population level or how individual interneurons contribute to shaping cell assemblies. To clarify the general role of interneurons in population dynamics and synchrony without parsing them into distinct subtypes, we employed an all-optical approach [36,37] in *GAD67-Cre* mice [38], enabling us to select interneurons across a broad spectrum within the CA1 *stratum pyramidale*. This experimental approach mirrored our computational modeling that differentiated neuronal populations into two broad categories, excitatory (E) and inhibitory (I) neurons. Specifically, we have combined two-photon calcium imaging in mice running spontaneously on a non-motorized treadmill [15,39] with holographic light stimulation [36,37] of single interneurons using a soma-targeted opsin [40]. Our findings reveal that interneurons in the CA1 region tend to support the simultaneous activity of pyramidal cells while preserving the functional structure of endogenous cell assemblies, as evidenced by pairwise correlations and network bursts detected as synchronous calcium events (SCEs) [15,39]. Our data and modeling suggest that this effect stems from disinhibitory mechanisms, leading to a prolonged decrease in global inhibition. This amplification differs from the feature-specific suppression observed in the visual cortex [41] and may reflect the unique nature of recurrent connectivity. Thus, our findings imply that the impact of individual interneurons within CA1 assemblies is influenced by their integration into complex inhibitory circuits, beyond simply their cell type diversity.

## Results

### Imaging the activity of interneurons and pyramidal cells in CA1 in vivo

In order to describe the contribution of GABAergic interneurons to local hippocampal dynamics in vivo, we expressed the genetically encoded calcium indicator GCaMP6m in both pyramidal cells and interneurons of the CA1 hippocampus (Fig 1A, see Methods). To identify interneurons, we expressed the red protein tdTomato in GAD67-expressing neurons using the

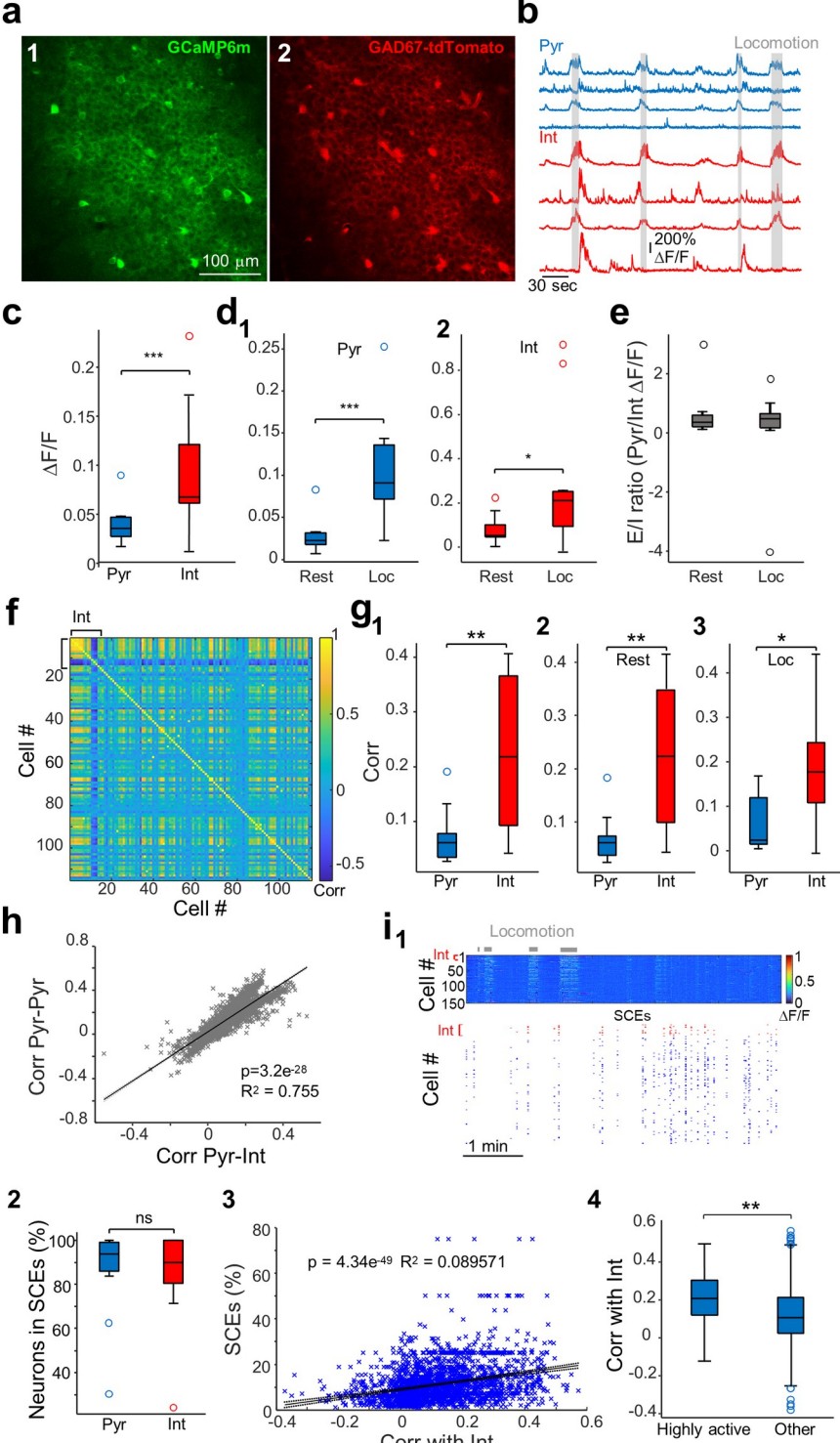

**Fig 1. CA1 interneuron activity is linked to pyramidal cell synchrony.** (**a**) GCaMP6m (1) and tdTomato (2)
fluorescence signals to—respectively—record neural activity and identify interneurons. TdTomato is expressed under
the control of the GABAergic promoter *GAD67*. (**b**) Example calcium ΔF/F traces from 4 pyramidal neurons
(tdTomato-negative) and 4 interneurons (tdTomato-positive). (**c**) Interneurons show higher variations in calcium
fluorescence signal than pyramidal cells ($p = 0.007$, Wilcoxon signed rank test, $n = 11$ FOVs from 6 mice). (**d**)
Pyramidal cells (d1) and interneurons (d2) display increased activity during locomotion (pyramidal cells: $p < 0.001$,
interneurons: $p = 0.041$, $n = 11$ FOVs from 6 mice, Wilcoxon signed rank tests). (**e**) E/I ratio (ratio between pyramidal

cells and interneuron ΔF/F) remains stable across rest and locomotion ($p = 0.8$, $n = 11$ FOVs from 6 mice, Wilcoxon signed rank test). (**f**) Correlation matrix for the recording shown in a (including all 15 interneurons and 100 pyramidal cells). (**g**) Pairwise Pearson's correlations between interneurons are higher than correlations between pyramidal cells. (**g1**) Whole recording: $p = 0.003$. (**g2**) Rest periods: $p = 0.002$. (**g3**) Locomotion: $p = 0.01$. All Wilcoxon signed rank tests, $n = 11$ FOVs from 6 mice. (**h**) Fit of linear model between pairwise Pearson's correlations of each pyramidal cell to interneurons and other pyramidal cells ($n = 2,793$ pyramidal cells, 11 FOVs, 6 mice). (**i1**) Example of SCEs occurring during rest. Locomotion periods are marked in gray. Top, heatmap shows the ΔF/F traces of all imaged neurons. The first rows correspond to interneurons (Int) and are indicated in red. Bottom, the raster plot shows the activity of imaged neurons during SCEs (see Methods for details on the detection method). Interneurons are highlighted in red and pyramidal cells in blue. (**i2**) Pyramidal cells and interneurons are recruited in SCEs in similar proportions ($p = 0.9$, Wilcoxon signed-rank test, $n = 11$ recordings from 6 mice). (**i3**) Fit of linear model between Pearson's correlations of each pyramidal cell to interneurons and percentage of SCEs that a cell participates to ($n = 2,793$ pyramidal cells, 11 recordings, 6 mice). (**i4**) Pyramidal cells that are highly active in SCEs (scoring above the 90th percentile in the distribution of SCE participation including all pyramidal cells) display higher Pearson's correlations to interneurons compared to other pyramidal cells (<90th percentile, $p = 1.1e^{-24}$, Mann–Whitney U test, highly active pyramidal cells: $n = 276$; other pyramidal cells: $n = 2,517$). * $p < 0.05$; ** $p < 0.01$; *** $p < 0.001$. Boxplots represent medians (center) and interquartile ranges (bounds). The whiskers extend to the most extreme data points not considered outliers, which are plotted individually using the circles. See also S1 and S2 Figs. Underlying data can be found in S1 Data. FOV, field of view; SCE, synchronous calcium event.

*GAD67-Cre* mouse line [38] (Fig 1A). This mouse line allows for a large sampling from diverse interneuron subtypes in the CA1 pyramidal layer, including the most representative ones [23], the PV-expressing basket and bistratified cells [1], the NOS-expressing ivy cells [42] and 2 types of interneuron-selective interneurons (ISI 1 and 3) that express calretinin [1,28] (see S1 Fig and Methods for quantification). A chronic glass window was implanted just above the dorsal hippocampus to image the calcium dynamics of CA1 pyramidal cells and interneurons using two-photon microscopy. Mice were head-fixed and free to run in the dark on a non-motorized treadmill allowing spontaneous movement in conditions minimizing external sensory influences.

In these conditions, CA1 dynamics are organized into sequences of neuronal firing during run and SCEs during rest periods [15,39]. These SCEs often co-occur with SWRs and activate functionally orthogonal cell assemblies, as previously shown [15]. We focused on the *stratum pyramidale* of the CA1 region where interneurons can be imaged together with excitatory principal cells. Each field of view (FOV) allowed for the simultaneous imaging of 254 ± 104 pyramidal cells and 11 ± 3 interneurons (means ± standard deviations; ranges: 115 to 402 and 8 to 15, respectively, $n = 11$ FOVs from 6 mice). In line with the classical observation of interneurons displaying higher firing rates than pyramidal cells [43,44], we found that interneurons showed higher amplitude changes in their calcium fluorescence signal from baseline compared to pyramidal cells ($p = 0.007$, Wilcoxon signed rank test, $n = 11$ FOVs from 6 mice, Figs 1C and S2a). Although calcium fluorescence signals are unlikely to report all the spikes fired by an individual neuron, this suggests that our experiments captured an accurate read-out of pyramidal cell and interneuron physiology. Pyramidal cells and interneurons displayed similar behavioral state-dependent activity, with, on average, increased activity from rest to locomotion (pyramidal cells: $p < 0.001$, interneurons: $p = 0.041$, $n = 11$ FOVs from 6 mice, Wilcoxon signed rank tests, Fig 1D). In line with this, the excitation to inhibition ratio (E/I, see Methods) did not change between rest and locomotion ($p = 0.8$, $n = 11$ FOVs from 6 mice, Wilcoxon signed rank test, Fig 1E). Next, we asked to what extent interneurons displayed conjoint activity. We found that pairwise Pearson's correlations between interneurons were higher than correlations between pyramidal cells (Fig 1F and 1G, whole recording: $p = 0.003$), and this difference was not behavioral state-dependent (rest periods: $p = 0.002$; locomotion: $p = 0.01$, all Wilcoxon signed rank tests, $n = 11$ FOVs from 6 mice). To control for potential biases in the correlation results due to different ΔF/Fs between the 2 populations, we subsampled

pyramidal cells to match interneurons' ΔF/Fs in each recording (see Methods for details). In these conditions, interneurons still showed significantly higher pairwise correlations ($p$ = 0.042, $n$ = 11 FOVs from 6 mice, Wilcoxon signed rank test, S1c Fig). This indicates that, although interneurons show higher morpho-physiological diversity than pyramidal cells, their activity is organized even more strongly in units of co-active neurons.

## The activity of CA1 interneurons is linked to pyramidal cells co-activity

Next, we sought to understand whether interneurons orchestrate pyramidal cell co-activity. We found that the more a single pyramidal cell was correlated to interneurons (on average), the more it was also correlated to other excitatory neurons (pairwise Pearson's correlations; linear regression: $p$ = $3.2e^{-28}$, $R^2$ = 0.755, $n$ = 2,793 pyramidal cells, 11 FOVs, 6 mice, Figs 1H and S2c), suggesting that interneuron activity could either just balance or even promote the synchronous recruitment of pyramidal cells. To extend this observation, we examined interneuron activity in relation to SCEs occurring during rest [15] (Fig 1I). SCEs occurred at a frequency of 0.06 ± 0.04 Hz ($n$ = 11 FOVs from 6 mice), a rate similar to previous observations and to the frequency of sharp-wave ripples (SWRs) in the awake state [15]. We observed that interneurons were present in the vast majority of SCEs and that the proportional recruitment of pyramidal cells and interneurons in SCEs was similar ($p$ = 0.9, Wilcoxon signed-rank test, $n$ = 11 FOVs from 6 mice, Figs 1I and S2d). Additionally, Pearson's correlations of individual pyramidal cells with interneurons positively predicted the proportion of SCEs pyramidal cells participated at (linear regression: $p$ = $4^{e-49}$, $R^2$ = 0.089, $n$ = 2,793 pyramidal cells, 11 FOVs, 6 mice, Figs 1I and S2e). In line with this, pyramidal cells that were highly recruited in SCEs (scoring above the 90th percentile in the distribution of participation including all pyramidal cells from all recordings) showed significantly higher pairwise Pearson's correlations with interneurons compared to other pyramidal cells (≤90th percentile, $p$ = $1.1e^{-24}$, Mann–Whitney U test, highly active pyramidal cells: $n$ = 276; other pyramidal cells: $n$ = 2,517, Fig 1I). Since pyramidal cells highly active in SCEs had significantly higher ΔF/Fs than other pyramidal cells ($p$ = 0.008, Mann–Whitney U test, $n$ = 276 highly active cells, $n$ = 2,517 other cells, from 11 FOVs from 6 mice), we controlled for potential bias due to higher spontaneous activity by sub-sampling less active pyramidal cells to match the highly active ones (see Methods). We still observed a significantly higher correlation with interneurons for the population highly active in SCEs ($p$ = $9.7e^{-14}$, Mann–Whitney U test, $n$ = 276 highly active cells, $n$ = 275 other cells, from 11 FOVs from 6 mice).

We conclude that the CA1 region hosts a balanced network both at the population and single-cell levels, with interneurons being more functionally connected than their excitatory partners. In addition, the more a cell displayed coordinated firing with interneurons, the more likely it was to be recruited within SCEs. Altogether, this suggests a contribution of interneurons to the coordination of activity in the CA1 region.

## All-optical interrogation of CA1 interneurons in vivo

In order to go beyond these observations and probe causality, we next tested the network influence of single interneurons using holographic photostimulation of an excitatory opsin [45,46] combined with calcium imaging. With this aim, we expressed the fast soma-targeted opsin ST-ChroME in GAD67-expressing CA1 neurons (interneurons) and GCaMP6m in all neurons (interneurons and pyramidal cells, Fig 2C and Methods). For simultaneous calcium imaging and photostimulation, we used a custom-built set-up with 2 LASER sources (920 and 1,030 nm) for both imaging and optogenetic activation (Fig 2A, see Methods). Mice were head-fixed and imaged in the same conditions as above. They spontaneously alternated between run and

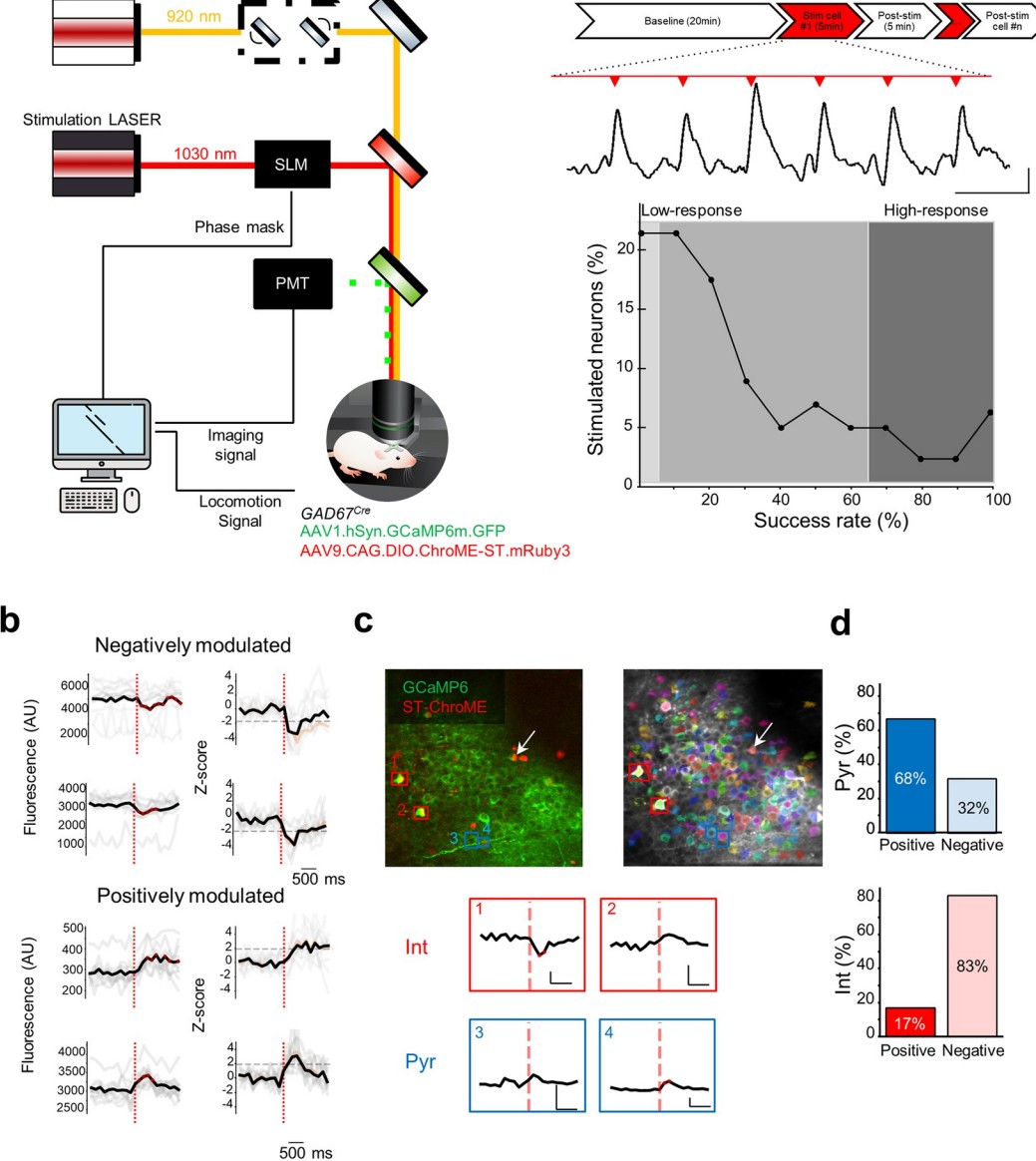

**Fig 2. Interneuron and pyramidal cell responses to single interneuron photoactivation.** (**a1**) Schematic representation of the custom-built optical set-up for targeted single-cell activation using holographic photostimulation combined with calcium imaging. Two LASER sources were employed for imaging and stimulation (920 and 1,030 nm, respectively). The fast soma-targeted opsin ST-ChroME was co-expressed with GCaMP6m in GAD67-Cre mice. Mice were head-fixed and free to run on a self-paced treadmill. SLM, spatial light modulator; PMT, PhotoMultiplier Tube. (**a2**) Schematic of the experimental timeline with the fluorescence calcium trace of a stimulated interneuron (Stim cell) during the stimulation epoch. The graph represents the distribution of success rate (i.e., fraction of stimulation trials inducing a significant calcium fluorescence response) among the targeted interneurons co-expressing ST-Chrome and GCaMP6m (149 interneurons, 11 mice). (**b**) Representative example fluorescence traces (left) of 4 imaged neurons negatively (top) or positively (bottom) modulated by the photoactivation of a single interneuron. The z-score (right) of their response was used to define significantly modulated neurons. The dashed red line indicates the time of stimulation. Single trials are indicated in light gray and the median trace in black. (**c**) Left, representative field of view of the CA1 region imaged in a head-fixed mouse in vivo. GCaMP6m (green) is expressed in all neurons, whereas ST-Chrome (red) is present only in GABAergic neurons. The photo-stimulated cell is indicated by a white arrow. Bottom, example stimulation-triggered fluorescence traces of non-stimulated interneurons (red) and pyramidal cells (blue). Right image indicates the segmented contour map of imaged neurons using Suite2P[88]. Scale bar: x: 500 ms, y: 5% fluorescence signal. (**d**) Bar plots of the distribution of positively (left)- vs. negatively (right)-modulated neurons among the pyramidal cells (top, blue) and interneurons (bottom, red). See also S3 Fig. Underlying data can be found in S2 Data.

rest periods ($n$ = 11 mice). After a baseline epoch of calcium imaging lasting 20 min, a single interneuron was targeted in the FOV for photoactivation by a train of light pulses every 30 s. The stimulation period lasted 5 min and was followed by a 5-min-long recovery period (i.e., no stimulation, Fig 2A). In this way, each targeted neuron was stimulated 10 times per stimulation period. A total of 149 interneurons from 53 FOVs were stimulated. We first quantified how efficient the stimulation was per targeted cell by computing the fraction of the stimulation trials that induced a significant calcium response (see Methods). When considering all the trials from all the neurons, we found that about a third of the trials induced a significant response (433 out of 1,490 trials). This rate is comparable to previous reports [45]. However, the stimulation efficiency was not evenly distributed among targeted interneurons (Fig 2A). Indeed, a few of them responded to less than 10% of the trials (21%, $n$ = 32 "low-response" cells) while only a minority (15%) were reliably entrained by the stimulation and responded to more than 60% of the trials ($n$ = 22 "high-response" cells). The remaining majority (64%) responded to between 10% and 60% of the trials ($n$ = 95 "medium-response" cells). Of note, cells displaying higher baseline activity exhibited a greater success rate (Pearson r = 0.206, $p$ = 0.012, S3c Fig). We also performed control experiments in mice for which interneurons expressed tdTomato instead of the opsin and found an average success rate of 8% of the trials (15 out of 190 trials, 19 targeted cells, 7 mice). With these control experiments, we can conclude that "low-response" (less than 10% of the trials) cells can also be defined as "unresponsive." The distribution of experiments and modulated cells per animal can be found in S1 Table.

## Optical activation of single interneurons differentially modulates the activity of other interneurons and glutamatergic cells

We next asked whether the stimulation of a single interneuron could in turn induce a significant modulation of the activity of other imaged cells. To this end, we examined all the non-stimulated neurons and compared their calcium fluorescence between a window of one second prior to stimulation and a window of one second after stimulation. We used a Z-score-based method to identify significantly modulated cells (see Methods). We found that the activity of a small subset of imaged cells (62 neurons from 23 FOVs in 6 mice) could be identified as positively or negatively modulated by the stimulation of single interneurons (Fig 2B–2D). There was a significant correlation between the number of neurons displaying indirect positive or negative modulation ($n$ = 39 and 23, respectively) and the fraction of successful stimulation trials in the target cell (Pearson's r = 0.251, $p$ = 0.002, Pearson's r = 0.295, $p$ = 0.0003, respectively, S3d Fig). No cells were indirectly modulated in the control experiments (19 targeted control cells, 7 mice) and only 1 neuron was detected as indirectly positively modulated in the "unresponsive cells" experiments (32 unresponsive neurons, 7 mice). Single-cell influence mapping experiments performed in the visual cortex indicated a preexisting functional relationship between the stimulated neuron and the cells indirectly modulated by the stimulation [41]. To test whether this was the case for single interneuron stimulation in CA1, we computed the baseline Pearson correlations between pairs of stimulated and indirectly modulated cells. Our results showed that both positively ($n$ = 39) and negatively ($n$ = 23) modulated cells tended to display a higher correlation with the stimulated neuron during baseline compared to unmodulated cells, but the difference was not statistically significant (Kruskal–Wallis H-test, 3 groups, $p$ = 0.17, see S3b Fig). The anatomical distance from the stimulated interneuron was not significantly different between modulated and unmodulated cells (Kruskal–Wallis H-test, 3 groups, $p$ = 0.19, S3a Fig).

A further question was whether directly modulated neurons were glutamatergic or GABAergic. We found that positively modulated cells were evenly distributed among

pyramidal cells and interneurons (0.13% and 0.11%, Z = −0.16, $p$ = 0.87, Z-test), but interneurons tended to be more negatively modulated than pyramidal cells (0.07% and 0.64%, Z = 1.87, $p$ = 0.06, Z-test, Fig 2D). We observed different trends in cell modulation within 2 populations, as determined by Barnard's exact test ($p$ = 0.038). Among the excitatory cell population ($n$ = 56 neurons), cells were more positively (68%) than negatively (32%) modulated. In contrast, the very few interneurons that were indirectly modulated were mostly inhibited (83%, $n$ = 6 cells). This imbalance could lead to a possible disinhibitory network effect. Accordingly, we found a significant decrease of 17 ± 35% (see Methods, $n$ = 734 interneurons, 11 mice) in global inhibition during the stimulation period as compared with the baseline ($p$ = 0.0023, unpaired $t$ test).

## Network modeling reveals that neuronal assemblies and dense but unspecific inhibition are required for the effects of single interneuron stimulation

To gain further insights into the circuit mechanisms underlying the functional integration of interneurons into local CA1 circuits, we simulated neuronal networks with global connectivity patterns similar to CA1. Our network model was composed of NE = 1,000 excitatory (E) and NI = 100 inhibitory (I) neurons, with sparse connectivity between E neurons (EE: 1%). Connections between E and I neurons, and specifically within I neurons, were dense (EI: 50% and II: 85%), in keeping with our experimental results (high I-I correlations in Fig 1G).

In addition, we implemented a variable subnetwork structure, whereby neurons belonging to the same subnetwork had stronger weights, forming cell assemblies. This was parameterized by a weight modulation factor (m), where m = 1 corresponds to highly specific subnetworks and m = 0 represents random connections without specificity (see Methods). Based on previous results on the presence of assemblies in CA1 [14,15,47], we chose mEE = 1 and mEI = 1. We chose non-specific connectivity for I-I connections (mII = 0), based on our experimental results (Fig 1G).

We then tested the effect of stimulating single interneurons in the model on the activity of other neurons (Fig 3A). We assessed what fraction of E and I neurons decreased or increased their activity as a result of stimulating all interneurons (Fig 3A). Inhibitory neurons were mainly suppressed, while the effect on excitatory neurons was diverse. Some E neurons decreased their activity, but a larger fraction of E neurons in fact increased their activity, presumably because of effective disinhibition in the network. Overall, the results corroborated and strengthened our experimental findings based on low numbers of indirectly modulated neurons (Figs 2D and 3A).

The large-scale rate-based network models can be analyzed in terms of their responses in the steady state [48], giving us the possibility to predict the results from the weight matrix directly. We performed such analysis and obtained the same results (S4A Fig), confirming that the effects described above arise from the connectivity structure. We then used this analytical insight to investigate the main ingredients of the connectivity which led to the differential effects of single interneuron stimulation on E and I neurons. We found that both E-I subnetwork structure and denser and less specific I-I connectivity were important for these results. First, in network models with no subnetwork structure (mEE = 0 and mEI = 0, Fig 3B), equal fractions of E neurons showed increase or decrease in their activity upon single interneuron stimulation, although I neurons were mainly suppressed (due to higher density of I-I connections). Second, when we allowed for a similar subnetwork structure for I-I connections with the same connection density (i.e., mII = 1 and II = 50%) as E-I connections, E and I neurons showed similar behavior, with slight dominance of positive changes (Fig 3C). The differential behavior could be retrieved when we made the I-I connections denser but still specific

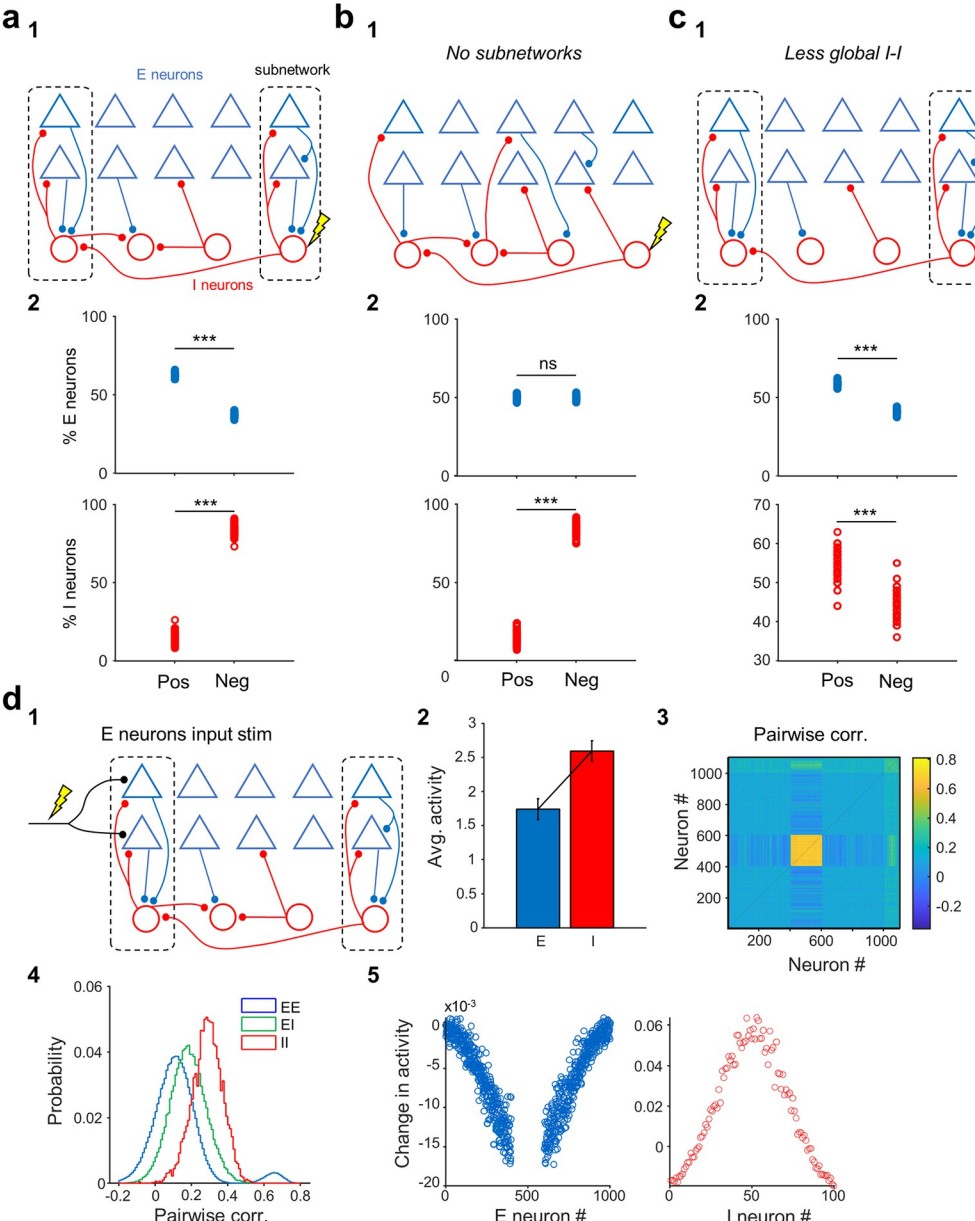

**Fig 3. Circuit mechanisms of the dichotomous effects of inhibition.** (**a1**) Schematic depicting the structure of connectivity in the network model. Excitatory (E) and inhibitory (I) neurons are organized in subnetwork structure, whereby neurons belonging to similar subnetworks have higher weights of connectivity. This is implemented by a subnetwork weight modulation factor, which is most specific for E-E and E-I synapses (m_EE = m_EI = m_IE = 1) and nonspecific for I-I connections (m_II = 0). Connection probabilities are different for different connection types, with E-E connections being drawn very sparsely (1%), E-to-I and I-to-E connections more densely (50%), and I-I connections the densest (80%) (see Methods for details). (**a2**) Fractions of E and I neurons showing a net positive or negative change in their activity, as a result of single I perturbations. Each datapoint denotes what fractions of excitatory (E) neurons (top) or inhibitory (I) neurons (bottom) are positively (Pos) or negatively (Neg) modulated upon perturbing a single I neuron (100 I neurons in total in the model). In these default connectivity parameters condition, E neurons have a shift towards positive modulation ($p = 6e^{-193}$, $n = 100$ I neurons), whereas I neurons have a negative bias ($p = 4e^{-194}$, $n = 100$ I neurons). (**b1**) Same as a1, when the structure of the network connectivity lacks subnetworks (m_EE = 0, m_EI = 0, m_IE = 0, m_II = 0). (**b2**) Differential modulation of E neurons is abolished without subnetworks (cell assemblies, $p = 0.11$, $n = 100$ I neurons), while the bias towards negative modulation in I neurons is maintained ($p = 6^{e-197}$, $n = 100$ I neurons). (**c1**) The structure of connectivity is the same as in (a1), but I-I connections have the same density (50%) and specificity as E-I connections (m_II = 1). (**c2**) Making I-I connectivity more specific led to positive shift for both E neurons ($p = 5e^{-191}$, $n = 100$ I neurons) and I neurons ($p = 3e^{-51}$, $n = 100$ I

neurons). (**d1**) Same as a1, but with consistent stimulation of the excitatory inputs onto 20% of E neurons of the network (see Methods for details). (**d2**) Average activity of E and I neurons in response to external stimulation depicted in d1. Error bars represent standard deviations. (**d3, d4**) Pairwise correlations between E (#1–1000) and I (#1001–1100) neurons in the network, following external stimulation. The distributions of correlations for different connection types (E-E, E-I, and I-E) are shown on the right. (**d5**) Changes in the activity of E (left) and I (right) neurons, as a result of stimulating a fraction of E neurons. The stimulated E neurons (in the middle) have a much higher increase in their activity as a result of direct stimulation and are hence not shown for illustration purposes. Neurons are organized according to their proximity in the subnetwork structure, namely close by neurons have a higher weight of connections, if their connection type is specific. Barplot in panel d2 represents means and standard deviations. See also S4 and S5 Figs. Underlying data can be found in S3 Data.

(mII = 1 and II = 85%), although suppression of I neurons was not as dominant as before (S4a Fig). We therefore conclude that both subnetwork structure within E-I connections and denser coupling of I-I connections are crucial to explain the result of our single interneuron stimulations in CA1. We assessed the statistical significance of our results by calculating the fractions of positively or negatively modulated postsynaptic neurons, as a result of each single inhibitory neuron perturbation (S5 Fig). Furthermore, the results of our single inhibitory neuron perturbations were robust to the choice of network parameters (S5 Fig).

To mimic synchronous SCE-like activity in our model network and cross-check our model and empirical data, we stimulated excitatory neurons with external inputs (Fig 3D). The external input was delivered in synchronous bouts to a fraction of excitatory neurons with functional proximity (i.e., close to each other in the subnetwork space) (see Methods for details), to emulate the input from CA3 to CA1. First, we observed average higher activity for I neurons, although E neurons were directly activated by the external stimulus (Fig 3D$_2$). These results were consistent with our experimental observations (Fig 1C). Higher activity of I neurons was a result of strong recurrent E-I connectivity in our network models, enabling a smaller number of I neurons to compensate for increases in the activity of more numerous E neurons.

Stimulating a fraction of E neurons led to a general suppression of activity in other E neurons (Fig 3D), including those close to the stimulated neurons in assembly structure (i.e., in subnetwork space). This was a result of strong and specific E-I connectivity in the network structure, as reflected in the strong and specific recruitment of I neurons (Fig 3D). These results argue for a general inhibition rather than competition between subnetworks. We also quantified pairwise correlations between E and I neurons and found correlations structures consistent with our experimental results (Fig 1G). On average, the correlations were highest between I-I pairs, intermediate for E-I pairs, and lowest for E-E pairs, although there was a wide distribution of E-E correlations (Fig 3D). The general structure of correlations was preserved when we stimulated I neurons instead of E neurons, although in terms of activity, inhibitory neurons were suppressed in this case (S4b Fig). E neurons showed a general inhibition, with neurons functionally closer to inhibitory neurons receiving more suppression, and some surround E neurons being disinhibited (S4b Fig).

### Interneuron activity is linked to pyramidal cell assemblies

Hence, in addition to the specific functional connectivity patterns of CA1, our simulations indicate that the presence of subnetwork structure in the form of mixed cell assemblies, comprising interneurons and pyramidal cells, is essential to explain our experimental observations. Thus, we next analyzed the contribution of interneurons to cell assemblies in our experimental data set. First, we detected the cell assemblies nested in SCEs using a k-means-based method [15]. We detected significant cell assemblies in 7/11 FOVs from 5 mice (2 ± 2 assemblies per FOV, mean ± standard deviation, range: 1–6 assemblies, Fig 4A). Consistent with a proportional representation of interneurons in SCEs, we found that the proportion of pyramidal cells

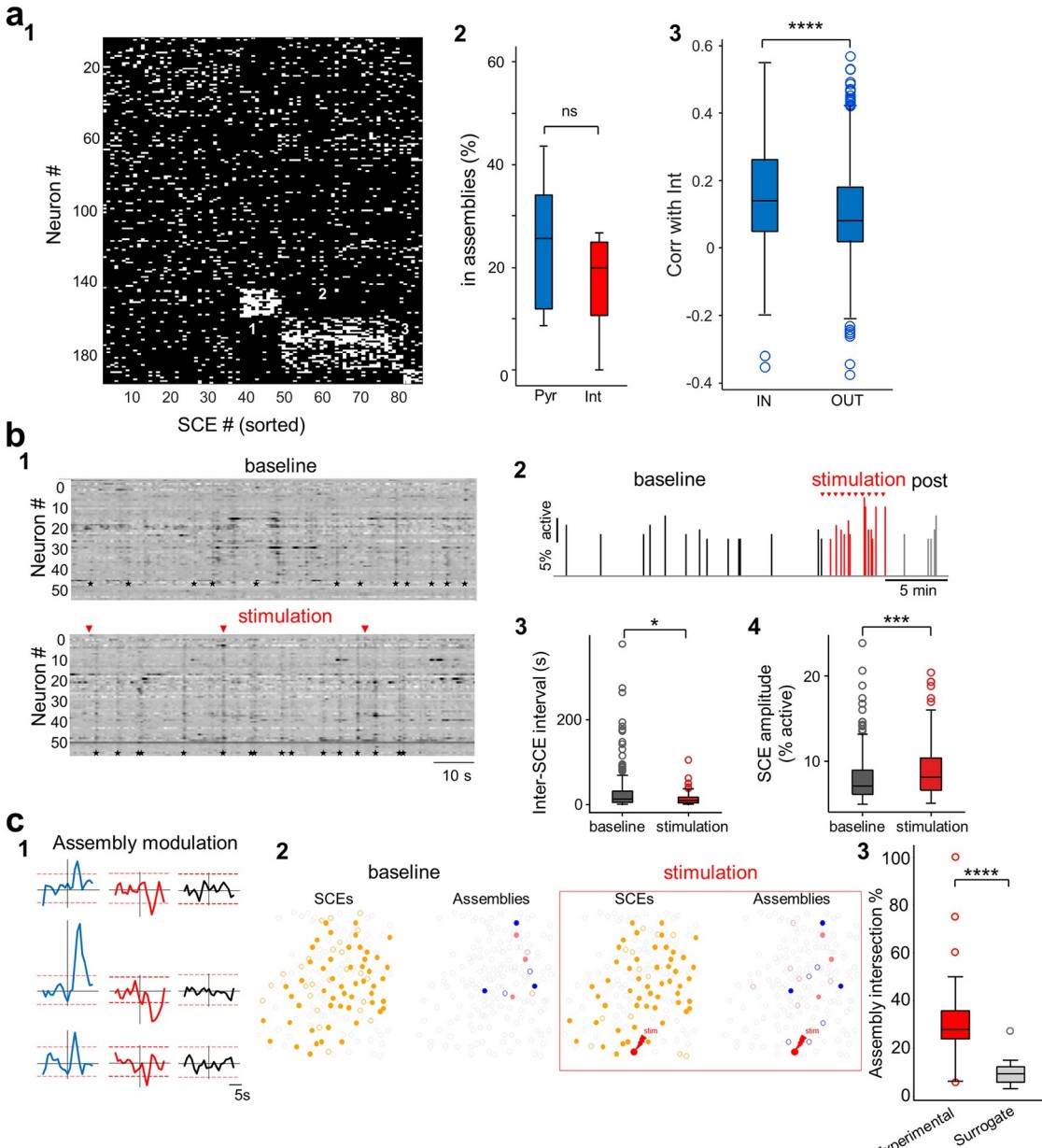

**Fig 4. Interneurons influence SCEs and pyramidal cell assemblies.** (**a1**) Raster plot of all SCEs, within 1 representative imaging session, sorted by cell assemblies detected using the k-means SCE method (see Methods for details). The 3 significant cell assemblies are numbered. (**a2**) Pyramidal cells and interneurons are recruited in cell assemblies in similar proportions ($p = 0.51$, Wilcoxon signed-rank test, $n = 11$ recordings from 6 mice). (**a3**) Pyramidal cells that are part of cell assemblies show higher Pearson's correlations to interneurons compared to pyramidal cells not forming cell assemblies ($p = 4e^{-11}$, Mann–Whitney U test, pyramidal cells in assemblies: $n = 49$; pyramidal cells not in assemblies: $n = 361$, 11 recordings, 6 mice). (**b1**) Raster heatmaps of the relative changes in fluorescence as a function of time for all active imaged neurons in a representative example, during baseline (top) and stimulation (bottom). Stimulation time points are indicated by red triangles. Note the increase in SCE (black stars) occurrence during stimulation **b2**. Histogram of the fraction of active cells in each detected SCE as a function of time during the course of an entire experiment. Note the increase in SCE amplitude and frequency during stimulation (red) **b3.** Box plots comparing the inter-SCE intervals during baseline and stimulation periods ($p = 0.01$, Mann–Whitney U test) for "high-response" cells (cf. with S7 Fig for all cells and unresponsive cells). (**b4**) Box plots comparing SCE amplitude (fraction of active neurons) during baseline and stimulation periods ($p = 0.007$, Mann–Whitney U test) for "high response" cells. (**c1**) Representative example traces of the median Z-score for members of the same assembly centered on the stimulation time. Assemblies are classified as activated (blue, left), suppressed (red, middle), or unmodulated (black, right) by the stimulation. Red dotted lines indicate significance thresholds (z-score>1,96 or <1,65, see Methods). (**c2**) Representative example showing cell participation in SCEs (orange) and 2 assemblies (blue and pink) during baseline (left) or stimulation and post-stim (right). Contours of neurons that are recruited in SCEs or that belong

to the same assembly in both epochs are filled. Open contours indicate cells that are only observed in a given condition. The stimulated cell is outlined in red. (**c3**) Cell assemblies recruited during the stimulation of single interneurons (experimental) show significantly more overlapping neurons with baseline-recruited assemblies compared to chance (surrogate) ($p = 2e^{-5}$, paired $t$ test, $n = 26$ assemblies from 20 FOVs from 3 mice). For each pair of the most similar assemblies between baseline and stimulation, the overlap of assembly members (intersections) was calculated between baseline and stimulation periods (experimental) or between baseline and a surrogate distribution of randomly drawn cells from neurons recruited in SCEs (surrogate).* $p < 0.05$; *** $p < 0.001$; **** $p < 0.0001$. Boxplots represent medians (center) and interquartile ranges (bounds). The whiskers extend to the most extreme data points not considered outliers, which are plotted individually using the circles. See also S7 and S8 Figs. Underlying data can be found in S4 Data. FOV, field of view; SCE, synchronous calcium event.

and interneurons forming cell assemblies was similar ($p = 0.51$, Wilcoxon signed-rank test, $n = 11$ FOVs from 6 mice, Figs 4A and S6). Additionally, we observed that pyramidal cells that were part of cell assemblies showed higher Pearson's correlations to interneurons compared to pyramidal cells not recruited into cell assemblies ($p = 4e^{-11}$, Mann–Whitney U test, pyramidal cells in assemblies: $n = 49$; pyramidal cells not in assemblies: $n = 361$, 11 FOVs, 6 mice, Fig 4A). Since pyramidal cells in assemblies had significantly higher ΔF/Fs than those not in assemblies ($p = 4.2e^{-6}$, Mann–Whitney U test, $n = 361$ in assemblies, $n = 1,437$ not in assemblies, from 11 FOVs from 6 mice), we controlled for potential bias due to higher spontaneous activity by subsampling less active pyramidal cells to match the highly active ones (see Methods). We still observed a significantly higher correlation with interneurons for the population forming cell assemblies ($p = 5.6e^{-6}$, Mann–Whitney U test ($n = 361$ in assemblies, $n = 319$ not in assemblies, from 11 FOVs from 6 mice).

Thus, interneurons either function in balancing the coordinated activation of pyramidal cells into assemblies or directly promote their recruitment into cell assemblies.

According to classical theories, hippocampal interneurons mainly operate in a feedback manner and segregate competing cell assemblies [11,49]. We wished to test whether our experiment revealed inhibition of competing pyramidal cells by interneurons. To this end, we examined the assembly activation-triggered average of pyramidal cells' calcium traces when each interneuron in an assembly was active or inactive (S6d Fig). Activity of competing assemblies or of pyramidal cells not forming assemblies was similar regardless of whether the interneuron in an assembly was active or inactive (S6d Fig). On average, the calcium transients of the same assembly were reduced in amplitude when the interneuron was inactive, but this effect was not statistically significant ($p = 0.7$, Wilcoxon signed-rank test, $n = 7$ FOVs with significant assemblies from 5 mice, S6d Fig). Thus, our data do not speak in favor of a segregating role of interneurons. Such empirical evidence is consistent with our model showing global inhibition rather than competition between subnetworks (Fig 3D).

To confirm that the observations on cell assemblies were not restricted to the detection method or were simply a reflection of the link between interneurons and SCEs, we employed a second cell assembly detection method not restricted to rest periods and SCEs. We used a procedure based on principal component analysis to detect significant assemblies, followed by independent component analysis to extract the weight of each neuron for each assembly (see Methods for details). We detected significant assemblies in all 11 FOVs, with $3 \pm 2$ assemblies per FOV (range: 1–6 assemblies). In keeping with a proportional contribution of pyramidal cells and interneurons to cell assemblies described previously, we observed similar assembly weights in pyramidal cells and interneurons ($p = 0.4$, Wilcoxon signed rank test, $n = 31$ assemblies from 11 FOVs from 6 mice, S6c Fig). In line with a positive relationship between interneurons and synchrony, we found that interneuron weights in a defined assembly predicted (positively) the weights of pyramidal cells ($p = 2.3e^{-5}$, $R^2 = 0.862$, S6c Fig, $n = 11$ FOVs, 31 assemblies, 6 mice). This indicates that higher interneuron participation is linked to stronger pyramidal cell co-activity. Thus, with both assembly methods, interneurons did not cluster

into a single assembly but appeared intermingled across all cell assemblies. Consistent with this, the pairwise Pearson's correlations between interneurons were not significantly higher than the correlations between interneurons and pyramidal cells ($p$ = 0.167, Wilcoxon signed rank test, $n$ = 11 FOVs from 6 mice, S6a Fig). Furthermore, pyramidal cell assembly weights were more clustered in a single assembly compared to interneurons (maximum assembly weight across assemblies divided by the absolute average of assembly weights; $p$ = 0.0025, Mann–Whitney U test, $n$ = 31 assemblies, from 11 FOVs from 6 mice, S6c Fig). This indicates that interneurons contribute to assembly activity in a more unspecific fashion. We conclude that interneurons contribute to cell assemblies proportionally to their representation within local circuits, potentially favoring their recruitment but, at least in our experimental conditions, not mediating cell assembly segregation.

## Causal involvement of single interneurons in SCEs

Since single interneuron activation results in an unbalanced local modulation of excitatory and inhibitory cells that could lead to an increased network excitability, we asked whether this impacted population dynamics and in particular the synchronous network bursts occurring during rest in the form of SCEs. SCEs occurred at a frequency similar to the previous experiments [15] (0.04 ± 0.02 Hz, $n$ = 24 FOVs from 5 mice). In the experiments with "medium-response" and "high-response" cells (10% and more responses), we observed a significant decrease in the inter-SCE intervals (Mann–Whitney U test, $p$ = 0.035) and an increase in the SCE amplitude (number of co-activated cells) (Mann–Whitney U test, $p$ = 0.0007) of SCEs during the stimulation period compared to baseline (18 FOVs, 5 mice, S7a Fig). Both of these changes were more prominent in the case of "highly-responsive" cells (more than 60% responses, Mann–Whitney U test, $p$ = 0.01 and $p$ = 0.007, respectively, Fig 4B). In contrast, "unresponsive" cells did not induce any significant change in the amplitude or frequency of SCEs (Mann–Whitney U test, $p$ = 0.45 and $p$ = 0.13, respectively, S7b Fig). Importantly, this change in network dynamics did not reflect a change in behavior since run epochs were similarly distributed during baseline and stimulation periods ($Z$ = −0.51, $p$ = 0.61, Z-test). We conclude that the stimulation of single interneurons enhances network synchrony in CA1.

## Single interneuron stimulation favors the reinstatement of endogenous assemblies

We last asked whether the network influence of single interneurons was modifying the endogenous functional structure of local circuits, including the organization of cell assemblies. To test whether the functional structure of our imaged network was affected by single interneuron stimulation, we first analyzed whether the cell assemblies composing SCEs (as defined in [15], see Methods) were affected. In total, we detected 96 assemblies during the baseline periods consisting of an average of 9 cells across 55 experimental sessions (out of total 149, from 11 mice). To determine the impact of stimulation on these assemblies, we first examined whether the stimulation evoked a significant time-locked response (activation or suppression) among cell assembly members. To this aim, we computed the average of the calcium fluorescence traces of cell assembly members within a 10-s time window centered on the time of the stimulation and used a Z-score-based test to determine whether the fluorescence signal just after the stimulation was significantly different from just before (see Methods). We found that about one-third of the assemblies (29 out of 96) were significantly modulated by single-interneuron stimulation (Fig 4C). The majority (58%) of the significantly modulated assemblies was activated following the stimulation (17 out of 29) while the rest were suppressed (12 out of 29). Therefore, single interneuron stimulation can lead to cell assembly activation or suppression.

Finally, we wished to assess whether interneuron stimulation favored the recruitment of endogenous cell assemblies or created new co-activity patterns. First, we compared the cell composition of SCEs between the baseline and stimulation periods (see Methods for details). We calculated the percentage of cells newly recruited in SCEs and compared the distributions of new cells between experiments with and without a significant activation of the stimulated interneuron. We found no significant difference between experiments for which the stimulated cell responded to more than 10% of the trials (i.e., "medium" and "high response cells") and experiments in unresponsive cells ($n = 70$ and 14 experiments with detected SCEs, respectively, $p = 0.43$, unpaired $t$ test). In the same way, there was no significant difference in SCE cell composition between "high-response" and "unresponsive" cell experiments ($n = 20$ and 14 experiments with detected SCEs, respectively, $p = 0.82$, unpaired $t$ test).

Last, we examined the impact of interneuron stimulation on cell assembly composition. To this aim, we determined the cell assembly composition during the baseline and compared it with the combined stimulation and poststimulation periods (in order to have enough SCEs to perform cell assembly clustering). For each pair of cells located within the same assembly during the stimulation and poststimulation periods, we determined the percentage that were also part of the same assembly during the baseline period (Fig 4C). We found a remarkable preservation of cell assembly membership since all of the pairs recruited in the same assembly during or just after stimulation were already part of a similar assembly during baseline (median 100%, $n = 236$ pairs, 3 mice). Consistent with this, the most similar assemblies between baseline and stimulation exhibited a significantly higher degree of overlap in member neurons compared to chance (Fig 4C$_3$, $p = 2e^{-5}$, paired $t$ test, $n = 26$ assemblies from 20 FOVs from 3 mice, see Methods for details). We conclude that single interneuron stimulation can suppress or activate cell assemblies while preserving the endogenous co-activity structure, as predicted in our model.

## Discussion

Using calcium imaging and holographic single-neuron photoactivation, we show that CA1 interneurons form a functionally connected network that promotes synchrony between pyramidal cells, especially in the form of SCEs occurring during awake immobility. Interneuron subcircuits operate within a balanced network where the activity of inhibitory cells matches that of excitatory ones across different behavioral states and at single neuron level. Optical activation of single interneurons triggers a transient disinhibition that favors network synchrony but does not alter the endogenous functional organization of cell assemblies. In agreement with our computational simulations, we propose that the ability of single interneurons to promote synchrony in CA1 results from such rigid modular organization in cell assemblies and the preferential interconnectivity among interneurons, rather than the intrinsic features of specific interneuron subtypes.

### Synchrony-promoting effects of inhibitory interneurons in CA1

Our results are consistent with empirical and theoretical work linking perisomatic interneurons to the generation of SWRs [34,50–53], which are characterized by significant concerted firing of pyramidal cells. We show that the more a single pyramidal cell correlates to interneuron activity, the more it correlates to other pyramidal cells, and it is recruited in SCEs and cell assemblies. Additionally, optical stimulation of single interneurons increased both frequency and amplitude of SCEs, and modulated cell assembly activity (in most cases enhancing recruitment of cell assemblies). In CA1, recurrent excitatory connectivity between pyramidal cells is low, and local interactions between pyramidal cells are primarily mediated by inhibitory interneurons [19]. Furthermore, approximately 20% of CA1 interneurons are interneuron-selective interneurons mediating disinhibition of pyramidal cells [54–56]. Tracing experiments

calculating the relative contribution of interneurons versus local principal cells to the synaptic inputs onto CA1 interneurons [18] are needed to see whether an imbalance comparable to the one reported in the dentate gyrus [57] applies to CA1. In any case, we have shown that interneurons, if indirectly modulated, are typically inhibited following stimulation of a single CA1 interneuron, speaking in favor of a disinhibitory effect through the dense recurrent inhibitory network in CA1, as supported by our simulations. It remains unclear whether these synchrony-promoting effects are a general feature of cortical networks or if they are favored by the anatomical connectivity of the CA1 region of the hippocampus. Similar single-cell optogenetic experiments have been performed in the upper layers of the visual cortex, where recurrent synapses are mostly glutamatergic [41]. In these conditions, the fine-tuning of inhibition relies on specific functional links between excitatory and inhibitory cells [58].

The increase in SCEs frequency and amplitude triggered by interneuron stimulation may also be mediated by other mechanisms than connectivity. One example could be depolarizing GABA [59]. Although this may seem implausible in the adult brain in vivo, recent evidence suggests that the reversal potential for GABA is more depolarized during prolonged wakefulness [60]. Another possibility that would combine both specific interneuron connectivity and intrinsic cell properties is the recently described persistent interruption of parvalbumin-expressing interneuron firing following brief inhibitory synaptic input [61]. Alternative options are the plasticity of interneuron synapses, gap junction coupling between interneurons, and long-range interactions with external structures such as the medial septum or the entorhinal cortex. Of note, gap junctions are frequently found among PV-basket cells found in the CA1 *stratum pyramidale* [62] but do not appear critically involved in the generation of network bursts in the form of SWRs [63]. In addition, the average distance between the somata of stimulated and indirectly modulated interneurons (137 ± 52 μm, not shown and S3A Fig) makes electrical coupling quite unlikely according to previous measures [64]. The increase in SCE frequency caused by single interneuron activation is likely to be indirect because SCEs were not locked to the stimulation (although in many cases cell assemblies were), and the proportion of positively modulated neurons was low. This could indicate that modulation of SCEs and cell assemblies may be caused by a slow buildup in the network, for example, mediated by a change in E/I. Future modeling work using biologically detailed models will be instrumental in suggesting specific mechanisms supporting single interneuron-mediated synchronization.

Our previous research demonstrated that hippocampal SWRs are frequently associated with SCEs [15]. In line with the interneuron modulation of SCEs, interneuron-pyramidal cell interactions have been shown to be crucial for SWRs [34,50]. In particular, stimulation of perisomatic interneurons of the CA3 pyramidal layer suppresses, and subsequently enhances, the generation of SWRs [53]. In this study, we reveal that CA1 network dynamics during rest, exemplified by SCEs, are also regulated by local inhibitory mechanisms. These mechanisms likely work in concert with excitatory inputs to CA1. Notably, CA3 and entorhinal cortex inputs provide feature-tuned excitatory signals to CA1 pyramidal cells during exploration [65–68], and CA3 inputs are associated with SCEs during rest [69]. Therefore, local interneurons might play a more active role in the reactivation of CA1 assemblies than previously recognized, complementing known excitatory inputs and dendritic clustering mechanisms [70,71].

## Going beyond interneuron subtypes

Our findings expand the functional repertoire of hippocampal interneurons in vivo. Thus far, many studies focused on pyramidal cell to interneuron functional connectivity, proposing that interneurons are feedback units recruited by pyramidal cell ensembles (place cells) to prevent runaway excitation or inhibit competing assemblies [11,25,26,49]. In contrast, our study puts a

spotlight on interneurons to pyramidal cell directionality, demonstrating that CA1 interneurons are powerful controllers of network dynamics, in agreement with recent optogenetic studies stimulating several interneurons from the same subtype [30,32,35,50]. Our study examined the CA1 pyramidal layer because this allows simultaneous monitoring of large ensembles of pyramidal cells and interneurons. Since a large proportion of interneurons in this layer is formed by basket cells and axo-axonic cells [23], it is reasonable to assume that the effects we described are mediated, at least in part, by perisomatic targeting interneurons. This is consistent with large unitary inhibitory responses in the local field potential observed when stimulating single basket cells [72,73] and with theoretical work showing that parvalbumin-expressing basket cells strengthen and stabilize pyramidal cell assemblies [74]. Other inhibitory cells that could play a role are interneuron-selective interneurons (which is in line with the prevalent inhibition of interneurons observed) [28,30] or dendrite-targeting ivy and bistratified cells [42,75]. However, ivy cells, axo-axonic and VIP-expressing interneurons are inhibited during ripples [30,76–79]. Importantly, results may be different when stimulating single interneurons in other layers, which are populated by different cell types. We hypothesize that distal dendrite-targeting interneurons (e.g., OLM cells) may serve more to gate tuft-initiated synaptic plasticity than to orthogonalize cell assemblies. Still, variability in the ability of a single interneuron to exert a strong network influence appeared to correlate mostly with the reliability of light entrainment rather than anything else (S3 Fig). Therefore, we would like to propose that single-interneuron influence is exerted by interneuron interconnectivity rather than specificity. This is somehow in agreement with a previous study putting forward an overall functional homogeneity of interneurons [18]. Future studies could clarify whether the variability in the effects of stimulation is influenced by cell type, or perhaps also by brain state. For example, stimulating an interneuron during rest, when activity is typically reduced, may produce different or more detectable responses in the network. In addition, this study was performed in conditions where the mice are not running towards any specific goal and deprived of external sensory influences. In these conditions, hippocampal dynamics are dominated by self-referenced information [39,80–82] and the influence of single interneurons in other contexts may be different.

## Implications and limitations

The behavioral implications of interneuron-mediated enhancement of synchrony remain to be established. Given the importance of SWRs and hippocampal reactivations for memory consolidation [47,83,84], one possibility is that CA1 interneurons of the pyramidal layer coordinate memory consolidation during rest by promoting network bursts. In line with this hypothesis, inhibition of parvalbumin-expressing interneurons in CA1 after learning impairs fear memory consolidation and pyramidal cell firing coherence [85].

This study reveals that the E/I ratio remains balanced across brain states, with CA1 interneurons recruited in SCEs and cell assemblies in similar proportions to pyramidal cells. This observation may seem at odds with previous results obtained using extracellular spikes recordings [86]. It could be explained by the bias towards sampling fast-spiking interneurons (e.g., parvalbumin-expressing basket cells) when using extracellular electrophysiological recordings. In addition, analysis of the distribution of individual cells' weights across assemblies showed that interneurons are less associated with a single assembly than pyramidal cells. This is consistent with what we previously observed for early-born inhibitory hub cells [29]. Our data indicate that interneurons tend to promote assembly activation, rather than only assembly segregation. We found no evidence of spontaneous assembly segregation by single interneurons. However, it is important to keep in mind that calcium imaging may not detect a certain proportion of spikes or be able to detect monosynaptic inhibition, particularly in the case of

neurons with low firing rates. Optical activation of single interneurons caused assembly activation in approximately 60% of the cases and inhibition in 40%. We also found that activating single interneurons triggered activation of previously active assemblies, rather than creating new associations between neurons. This observation provides further support to the concept that hippocampal dynamics are preconfigured by functional connectivity [87,88].

There are important limitations to consider when interpreting our results. First, calcium imaging has a low temporal resolution compared to electrophysiology. This implies that short-delay fast dynamics may be missed by our analyses (for example, transient inhibitions followed by excitation). Additionally, it is unclear how much calcium dynamics can disclose inhibitory responses. Thus, we cannot fully exclude that assembly segregation mediated by inhibition plays a stronger role when recording from CA1 neurons with high temporal resolution or from interneurons located outside the *stratum pyramidale*. However, we did find that most indirectly modulated interneurons displayed inhibitory responses. Finally, calcium imaging is unlikely to reveal all the spikes fired by CA1 neurons. The proportion of spikes revealed by calcium-related fluorescence is particularly uncertain when examining fast spiking populations. To the best of our knowledge, no study to date performed dual electrophysiology and calcium imaging recordings from interneuron populations to provide a convincing benchmark. Thus, we cannot exclude that heterogeneities in interneuron recruitment observed with calcium imaging may be due to inherent differences in the spike-to-calcium relationship for the different subtypes. Nonetheless, we found that interneurons, on average, display significantly higher $\Delta F/Fs$ than pyramidal cells, indicating that calcium imaging can detect their higher firing rates.

## Conclusion

We provide converging experimental and modeling evidence for the role of single interneurons in triggering synchrony and endogenous cell assembly activation. This is likely due to the close interconnectivity between interneurons in adult CA1. In developing cortical circuits, single interneurons appear to control synchrony in the opposite direction [36]. Thus, the present finding has broad and important implications, including in pathology, such as epilepsy, where spatial coding deficits are related to disrupted interneuron synchronization [89].

## Materials and methods

### Animals

All experimental procedures were approved by the French ethics committee (Ministère de l'Enseignement Supérieur, de la Recherche et de l'Innovation (MESRI); Comité d'éthique CEEA-014; APAFiS #28.506) and were conducted in agreement with the European Council Directive 86/609/EEC.

GAD67-Cre mice were kindly donated by Dr. Hannah Monyer (Heidelberg University). Ai14 reporter mice were purchased from Jackson Laboratories (B6;129S6-Gt(ROSA)26Sor[tm14(CAG-tdTomato)Hze/J], strain # 007908). Mice were bred and stored in an animal facility with room temperature (RT) and relative humidity maintained at 22 ± 1°C and 50 ± 20%, respectively. Mice were provided ad libitum access to water and food.

### GCaMP6m, tdTomato, and ST-ChroME expression

GCaMP6m expression was obtained by injection of a viral vector in the dorsal CA1 in adult mice or in the lateral ventricle in newborn pups at P0. tdTomato expression in GABAergic neurons was achieved by crossing GAD67-Cre mice with Ai14 reporter mice or by injecting a

floxed viral vector expressing tdTomato either in the dorsal CA1 in adult mice or in the lateral ventricle in newborn pups at P0.

For viral injections in the CA1 of adult mice, GAD67-Cre mice (8 to 12 weeks of age) were anesthetized using 1% to 3% isoflurane in oxygen. Analgesia was also provided with buprenorphine (Buprecare, 0.1 mg/kg). Lidocaine cream was applied before the incision for additional analgesia. Mice were fixed to a stereotaxic frame with a digital display console (Kopf, Model 940). Under aseptic conditions, an incision was made in the scalp, the skull was exposed, and a small craniotomy was drilled over the target brain region. A recombinant viral vector was delivered using a glass pipette pulled from borosilicate glass (3.5" 3-000-203-G/X, Drummond Scientific) and connected to a Nanoject III system (Drummond Scientific). The tip of the pipette was broken to achieve an opening with an internal diameter of 25 to 35 μm. To express GCaMP6m, AAV1.Syn-GCaMP6m (pAAV.Syn.GCaMP6m.WPRE.SV40 from Addgene, #100841, titer $6-8 \times 10^{12}$) was injected. To express tdTomato in GABAergic neurons, AAV9-FLEX-tdTomato was injected (pAAV-FLEX-tdTomato from Addgene, #28306, titer $3 \times 10^{12}$). Viruses were diluted in D-phosphate-buffered saline (PBS, Sigma Aldrich). To target the dorsal CA1, we injected 600 nL at a rate of 25 nL/min at the coordinates below. All coordinates are in millimeters. Anteroposterior (AP) coordinates are relative to bregma; mediolateral (ML) coordinates are relative to the sagittal suture; dorsoventral (DV) coordinates are measured from the brain surface. Dorsal CA1: −2 AP, −2 ML (300 nL at −1.3 DV and 300 nL at −1.25 DV).

For P0 injections, we followed previously published procedures [83,84]. Briefly, mouse pups were anesthetized by hypothermia. GAD67-Cre mouse pups were injected in the left hemisphere. To reach the ventricle, we injected in a position that was roughly two fifths of an imaginary line drawn between lambda and the left eye at a depth of 0.4 mm. Correct injection was verified by the spread of the blue dye. To express ST-ChroME in GABAergic neurons, AAV9.DIO-ST-ChroME-P2A-H2B-mRuby3 was used (pAAV-CAG-DIO-ChroME-ST-P2A-H2B-mRuby3 from Addgene, #108912, titer $2.7 \times 10^{12}$).

In order to verify that the targeted cells using GAD67-Cre mice in our imaging experiments were GABAergic cells, several immunohistochemical labeling experiments were performed using antibodies directed to parvalbumin (PV, Swant, PVG-214, AB_2313848), neuronal nitric oxide synthase (nNos, Sigma-Aldrich, n2280, AB_260754), calretinin (CR, Swant, CG1, AB_10000342), GAD67 (Merck, MAB5406, AB_2278725), and GABA (Sigma-Aldrich, A2052, AB_477652). Using GAD67-Cre/Ai14 mice, we quantified labeling in CA1 and found that virtually all neurons expressing PV, CR and nNOS were also expressing Td-Tomato (PV: 98.39%, $n = 3$ animals, $n = 222$ cells; CR: 100%, $n = 3$ animals, $n = 95$ cells; nNOS: 100%, $n = 3$ animals, $n = 73$ cells, S1A and S1D–S1F Fig). In addition, the vast majority of Td-Tomato neurons (90.47%, $n = 5$ animals, $n = 986$ cells) were immunopositive for GAD67 and/or GABA (S1A–S1C Fig). In addition, the distribution of ST-ChroME expressing cells corresponds to the known distribution of GABA neurons in CA1 (S1G Fig). In line with this, immunolabeling showed that ST-ChroME expressing cells were GAD67 positive (S1I–S1L Fig). We found no evidence of tdTomato or ST-Chrome expressing neurons with pyramidal-like structure (e.g., long apical dendrites reaching the lacunosum molecular), whereas the expression of GCaMP6m in neurites across the CA1 layers is consistent with expression in the vast majority of pyramidal cells (S1G Fig). These histological experiments validate the use of GAD67-Cre mice as robust models to tag GABAergic neurons in imaging experiments.

### In vivo 2-photon calcium imaging

A chronic cranial window was implanted using previously published procedures [15,29,39]. Mice were head-fixed on a non-motorized treadmill allowing self-paced locomotion. All

experiments were performed in the dark. No reward was given. After 3 to 5 habituation sessions, mice were alert but calm and alternated between periods of locomotion and rest during imaging. The treadmill was made of a 180-cm black velvet seamless belt lacking tactile or visual cues mounted on 2 wheels. The movement of the treadmill was monitored using 2 pairs of LEDs and photo-sensors that read patterns from a disk attached to one of the wheels. For all experiments, extra sound, odor, touch, and light were minimized during the imaging session. Imaging was performed with a single beam multiphoton laser scanning system coupled to a microscope (TriM Scope II, Lavision Biotech). The Ti: sapphire excitation laser (Chameleon Ultra II, Coherent) was operated at 920 nm for GCaMP6m excitation and at 1,030 nm for tdTomato excitation. Fluorescence emission was acquired using a 16× objective (Nikon, NA 0.8) and split in 2 detectors (GaSP PMT, H7422-40, Hamamatsu) with bandpass filters of 510/10 nm for GCaMP6m and 580/20 nm for tdTomato. Scanner and PMTs were controlled by a commercial software (Imspector, Lavision Biotech). To optimize the signal-to-noise ratio of fluorescence variation, we used a dwell time exposition of 1.85 μs and a spatial resolution of 2 μm/pixel that allowed us to acquire at 8 to 10 Hz at a field of view of 400 × 400 μm. Locomotion and imaging triggers were synchronously acquired and digitized using a 1440A Digidata (Axon instrument, 2 kHz sampling) and the pClamp 10 software (Molecular Devices).

## In vivo 2-photon calcium imaging with simultaneous holographic optogenetic stimulation

To record from the entire CA1 network while stimulating individual interneurons, we employed an all-optical approach that we recently employed to stimulate individual GABAergic neurons in the developing barrel cortex [36]. As in the experiments without holographic optogenetic stimulation, a chronic cranial window was implanted using previously published procedures [15,29,39]. Mice were head-fixed on a non-motorized treadmill allowing self-paced locomotion. All experiments were performed in the dark with no reward. After a brief habituation period, mice began to alternate between periods of locomotion and rest during imaging. The movement of the animal was recorded using Phenosys SpeedBelt treadmill. The optical system was a custom-built microscope combining galvo-based two-photon scanning with Computer Generated Holography [37,90,91]. Raster scanning of calcium fluorescence signals was achieved using standard galvo scanners and a pulsed femtosecond imaging LASER source. The LASER beam (Chameleon Ultra II, Coherent) was expanded with 2 lenses telescope assembly (f = 300 mm, f = 500 mm) and projected onto an XY galvo mirror pair (6215H, Cambridge Technology) controlled with 2 servo driver cards (67125H-1HP-FS60, Cambridge Technology). A half-wave plate (#AWHP10M-980, Thorlabs) and a polarizer (#GT10-B, Thorlabs) were used to adjust LASER power. Next, a scan and a tube lens (focal length fS = 50 mm and fT = 375 mm, respectively) were used to conjugate the XY scanner focal plane to the back focal plane of the microscope objective (16× Nikon, N.A 0.8). This configuration allowed scanning a field of view of 280 μm × 280 μm (256 pixels × 256 pixels) at the focal plane of the objective with a frame rate of 8.4 Hz and a power of 50 mW at 920 nm wavelength. To collect the emitted fluorescent signal, the back focal plane of the objective and the focal plane of a GaAsP PMT (Hamamatsu, H7244-20) were conjugated through a relay of lenses (f = 100 mm, #AC254-100-A, Thorlabs, f = 25 mm, #LA1951-A, Thorlabs). Two spectral filters were mounted in front of the PMT (FF01-770/SP-25, Semrock, ET520/40m, Chroma) to optimize GFP detection. The analog signal was next converted from current to voltage and amplified through a transimpedance amplifier (#TIA60, Thorlabs). Finally, an electronic card (NI6356, National Instruments) in combination with Scanimage software (Vidriotechnologies) was used to control the scanners and to digitalize the analog signal from the PMT.

Photostimulation of neurons of interest used Computer Generated Holography. Briefly, the beam of the pulsed femtosecond photoactivation LASER (GOJI, AMPLITUDE SYSTEMS, 10 MHz repetition rate, 1,030 nm) was shaped by a Spatial Light Modulator (Hamamatsu, LCOS-SLM X13138-07). The size of the LASER beam was expanded using a two-lens telescope assembly (#AC254-030-B, Thorlabs, #AC254-150-B, Thorlabs) so that it covered the entire surface of the SLM. A half-wave plate (#AHWP10M-980, Thorlabs) was used to align the polarization of the laser to the orientation of the liquid crystals. Three lenses (#AC508-300-B) combined with the tube lens (fT = 375 mm) in a 4-f configuration enabled conjugating of the SLM focal plane to the back focal plane of the microscope objective. The zero-order of the SLM was suppressed with a cylindrical lens (f = 300 mm, f = #LJ1558L1-B, Thorlabs) as described previously [92]. A custom software (Wavefront Designer IV) based on the Gerchberg–Saxton algorithm was used to convert the photostimulation intensity pattern at the focal plane into a photostimulation phase mask addressed to the SLM [45].

To combine the 2 imaging and photostimulation paths, a dichroic mirror (#DMPSP1000L, Thorlabs) was placed at the focal plane of the scan lens. The custom software mentioned above was used to adjust the spatial overlap of the photostimulation pattern with the imaging at 920 nm thanks to a rhodamine fluorescent sample that was bleached at 1,030 nm and imaged at 920 nm. To synchronize the paths, a MATLAB script defined a photostimulation temporal gate and sent a TTL signal, via the NI card described above, to an obturator (Vincent shutter instruments) placed in front of the photostimulation LASER source during the raster scanning for calcium imaging. Holographic stimulation of targeted cells was achieved with an excitation spot of approximately 15 μm lateral size, corresponding to an axial resolution of 20 μm. Trains of 5 consecutive pulses (75 ms period, 25% duty cycle, at 0.3 to 0.8 mW/μm$^2$ power) were applied every 30 s during the stimulation period. The lateral size of the excitation spot and the use of a soma-targeted opsin (ST-ChroME) reduces the chance of photostimulating the neurites of non-targeted neurons. Additionally, GABAergic neurons are sparse and scattered, thus the chances of having 2 nearby neurons expressing the opsin are extremely low. Altogether, this approach leads to extremely low chances of stimulating more than 1 neuron at once.

Each experimental session consisted of 20 min of baseline recording, followed by 5 min of selected cell stimulation and 5 min of poststimulation recording. If the FOV contained more than 1 cell expressing both GCaMP6m and ChroME, the next cell was targeted and stimulated for 5 min, followed by 5 min of poststimulation recording. On average, 3 cells per FOV were stimulated. The proportion of time spent running was not significantly different between successive stimulations (Kruskal–Wallis H-test, $p = 0.89$).

### Analysis of calcium imaging data

In vivo calcium movies were preprocessed using the Suite2p toolbox for Python [88]. Movies were motion-corrected using rigid and non-rigid registrations with a block size of approximately one fourth of the size of the FOV in pixels. Automatic cell detection was performed based on activity (tau: 1 ms, equivalent to the GCaMP6m time constant; cell diameter: 5 to 7 pixels). To ensure correct segmentation of somatic calcium activity, the automatic detection was manually refined by adding and removing regions of interest (ROIs) with visual inspection of mean, maximum projections, and correlation images, as well as fluorescence traces. Subsequent analyses were performed using custom-made MATLAB (Mathworks, R2022b) and Python scripts. Locomotion epochs were defined as time periods with deflections in the photo-sensors signal reading the treadmill movement. Rest epochs were defined as periods >200 ms without treadmill movement.

$\Delta$F/F was calculated using the formula:

$$\Delta F/F(i) = \frac{x(i) - F0(i)}{F0(i)},$$

where $F0(i)$ is the median value within a 60 s sliding window before the frame $i$. The E/I ratio was defined as the ratio between the average $\Delta$F/F of pyramidal cells and average $\Delta$F/F of interneurons.

To control for potential biases in pairwise correlations between populations due to imbalances in $\Delta$F/Fs, we subsampled the less active population to match the more active one. We constructed a histogram of the neurons' $\Delta$F/Fs for each recording (using 5 or 10 bins). For each bin, we randomly sampled a number of neurons from the less active population equal to the number of neurons in the same bin from the more active population, skipping bins where either of the 2 populations was missing. SCEs were detected using a previously published method [15]. A third-order Savitzky–Golay filter with a frame size of 500 ms was first applied on the fluorescence calcium signal of each cell. The threshold for detecting calcium transients was adapted for each time point and each cell as follows: it was the sum of the median value with 3 times the interquartile range calculated within a −2/+2 s sliding window. To avoid detecting twice the same calcium transient, the minimal delay between events was set to 1 s. Activity occurring during run epochs was not included in this analysis. SCEs corresponded to the synchronous calcium events that involved more cells than expected by chance within a 200 ms time window (i.e., >3 standard deviations after temporal reshuffling of cell activity) and with a minimum cell number equivalent to 5% of the cells in the FOV.

We used 2 different cell assemblies detection methods. The first one was based on SCEs and k-means clustering. The second on principal component analysis (PCA) and independent component analysis (ICA). For the first method, cell assemblies were identified using a clustering algorithm based on SCE similarity for cell participation followed by a statistical test for cell participation in each SCE cluster. The SCE similarity metric was the squared Euclidean distance between columns of the normalized covariance matrix. This similarity metric allowed a more efficient clustering. Unsupervised clustering of SCE was obtained by running the k-means algorithm on this metric with cluster numbers ranging from 2 to 19. Hundred iterations of k-means were run for each cluster number and the iteration that resulted in the best average silhouette value was kept. For a given element $i$, the silhouette value was computed using the following formula:

$$s = \frac{b - a}{max\{a, b\}},$$

where $a$ is the average dissimilarity of $i$ with all other elements in its cluster and $b$ the lowest average dissimilarity of $i$ to any other cluster. In this analysis, the dissimilarity metric was the normalized covariance. A random distribution of average silhouette values for each cluster was calculated by reshuffling cell participation across different SCE and applying the same algorithm. Clusters with average silhouette values exceeding the 95th percentile of the random case were considered as statistically significant. Each cluster of SCE was then associated with a cell assembly which comprised those cells that significantly participated in the SCE events within that particular cluster. Cell participation to a given cluster was considered statistically significant if the fraction of SCE in that cluster that activated the cell exceeded the 95th percentile of reshuffled data. If a cell was significantly active in more than one SCE cluster, it was associated with the one in which it participated the most (percentage wise).

The second method is a PCA/ICA algorithm extensively used to detect hippocampal cell assemblies in electrophysiological data [93]. Fluorescence traces were convolved with a Gaussian kernel and then Z-scored (to reduce the influence of baseline fluorescence). The number of significant co-activation patterns (assemblies) was estimated as the number of principal component variances above a threshold derived from the circularly shifted matrix including the fluorescence traces of all neurons. Assembly patterns (vectors including assembly weights of all individual neurons) were then extracted with ICA.

## Analysis of all-optical data

In order to determine whether a stimulation (i.e., a 5-pulse train) evoked a significant calcium response on the target neuron, a dependent $t$ test was used to compare the values of the cell's raw fluorescence calcium signal 10 frames before (i.e., 1.2 s) and 10 frames after the stimulation. The stimulation was considered successful if the values after the stimulation were significantly ($p < 0.05$) higher than before stimulation.

A Z-score test was used to determine whether cells were indirectly modulated by the stimulation of a target neuron (activated or suppressed). For each stimulation, the fluorescence signal within a time window of 20 frames (i.e., 2.4 s) centered on the time of stimulation was considered. The median value for each time point within this interval was calculated. For the resulting trace, we calculated the Z-score, using the formula:

$$Z = \frac{x - \bar{x}}{\sigma},$$

both mean $\bar{x}$ and standard deviation $\sigma$ were calculated using the values before the stimulation. Modulated cells were selected using the following criteria: if the Z-score exceeded a value of 1.96 (95% confidence level) for 2 consecutive time points, a cell was defined as positively modulated; if it dropped below −1.65 (90% confidence level) for 2 consecutive time points, it was defined as negatively modulated. We chose different Z-score thresholds for positively and negatively modulated cells to account for calcium imaging's difficulty in detecting activity suppression (see [94]). Assembly modulation was analyzed using the same Z-score-based test as above but using the mean fluorescence of all the cells in the assembly.

Changes in global inhibition were estimated using the ΔF/F of the calcium fluorescence traces of interneurons. To calculate the percentage of change, we found the median difference between the ΔF/F values of the baseline period and the stimulation period. We also performed a dependent $t$ test on the data from the same periods to determine the significance of this change.

To compare SCE composition prior and during interneuron stimulation, we identified the subset of cells in the FOV that participated in at least 1 SCE during baseline. We then calculated the percentage of new cells appearing in the stim-poststim period compared to the baseline. Since the baseline period was twice as long as the stim-poststim period, we employed bootstrapping to account for the difference in duration. We randomly selected the minimum number of SCEs present in both periods 1,000 times. For each iteration, we calculated the percentage of new cells. Finally, we averaged the results across all iterations. Using this approach, we compared the distributions of new cells between experiments with and without a stimulation response.

To statistically determine if single interneuron stimulation favored the recruitment of endogenous assemblies, we employed the following method. First, we identified pairs of assemblies with the highest overlap of member neurons (intersection) between baseline and stimulation. For each baseline-stimulation pair of assemblies, we calculated the percentage of

overlapping cells. We then compared this to a surrogate distribution generated by randomly drawing, 1,000 times, the same number of cells as in the baseline assembly from those active during SCEs, calculating the intersection with baseline for each iteration, and averaging across the 1,000 iterations.

## Network modeling of neuronal responses

We simulate the activity of a network of $N$ neurons composed of $N_E$ excitatory and $N_I$ inhibitory units ($N = N_E + N_I$). The dynamics of neuronal activity is simulated by solving the following differential equations:

$$\tau \, dr_E(t)/dt = -r_E(t) + \Phi[W_{EE}r_E(t) + W_{EI}r_I(t) + s_E(t)] \tag{1}$$

$$\tau \, dr_I(t)/dt = -r_I(t) + \Phi[W_{IE}r_E(t) + W_{II}r_I(t) + s_I(t)],$$

which describe changes in the average firing rate of neurons as a function of external and recurrent inputs to them. Here, $r_E(t)$ and $r_I(t)$ are vectors of firing rates, composed of the activity of excitatory (E) and inhibitory (I) subpopulations at each time point, $t$. $\tau$ is the time constant of the network integration, and $\Phi(.)$ denotes the activation function, which is assumed to be a linear rectified function, namely: $\Phi(x) = 0$, if $x < 0$; $\Phi(x) = x$, if $x > 0$.

$s_E(t)$ and $s_E(t)$ denote the vectors of external inputs to E and I neurons, respectively, at each time point. All neurons receive a background input, which is modulated upon external perturbations or stimulation (this is described in detail in the section "External perturbation and stimulation"). The background input ($s_b$) consists of a mean component ($\mu_b$) and a noise term ($\zeta$): $s_b = \mu_b + \zeta$, where the noise term is drawn from a uniform distribution between $[0, \zeta_{max}]$ at each time step of simulation ($dt$). The equations are numerically solved by the forward Euler method.

Recurrent interactions between neurons are described by the weight matrix, $W$, with specific submatrices $W_{YX}$ describing the connection weights from a presynaptic subpopulation $X$ (E or I) to the postsynaptic subpopulation $Y$ (E or I). We describe how these weight matrices are obtained in the following section.

Unless stated otherwise, default parameters are chosen as: $N_E = 1000$, $N_I = 100$, $\tau = 10$, $dt = 1$, $\mu_b = 1$, $\zeta_{max} = 4$.

Connections between pairs of neurons from a presynaptic population $X$ to a postsynaptic population $Y$ are established based on the density of connectivity ($\epsilon_{YX}$). The connections are drawn from a binomial distribution with probability $\epsilon_{YX}$, returning a connectivity matrix $C_{YX}$ with 0 (no connection) or 1 (connected) entries. Self-connections are not permitted. Connections are assumed to be very sparse for E-E connections; for example, $\epsilon_{EE} = 0.01$ means that an E neuron is connected to 1% of other E neurons, on average. Other connection types are more densely established; for example, $\epsilon_{IE} = 0.5$ means that, on average, 50% presynaptic E neurons are connected to a postsynaptic I neuron.

The strength of the established connections is determined by the parameter $J$. For a given submatrix, $W_{YX}$, $J_{YX}$ denotes the average strength of connections. Each entry, $w_{ij}$ (specifying the weight of connection from the $j$-th presynaptic neuron to the $i$-th postsynaptic neuron), is obtained as follows:

$$w_{ij} = J_{YX}c_{ij}, \tag{2}$$

where $c_{ij}$ is the corresponding entry in the connectivity matrix denoting the presence (1) or absence (0) of a connection.

For nonspecific weight matrices ($m = 0$), $J_{YX}$ is the same for all pairs of neurons (belonging to the presynaptic subpopulation $X$ and postsynaptic population $Y$). The presence of subnetwork structure in CA1 is quantified by various degrees of specificity in different submatrices, with $m_{YX} = 1$ denoting the maximum specificity of connections weights from $X{\rightarrow}Y$, and $m_{YX} = 0$ recovering nonspecific weights. In either case, parameter $m$ quantifies the modulation of weights between a pair of neurons according to their proximity in a functional space. This is emulated by arranging E and I neurons on a ring, with parameter $\theta$ (ranging from $[0, \pi)$) specifying their location. The weight of a given pair of neurons is then obtained as follows:

$$w_{ij} = J_{YX}(1 + m_{YX}\, cos[2(\theta_i - \theta_j)])\, c_{ij}, \tag{3}$$

where $\theta_i$ and $\theta_j$ refer to the location of the $i$-th and the $j$-th neurons, respectively. E neurons are arranged to cover the range $[0, \pi)$ uniformly, such that the location of the $k$-th neuron is given by $\theta_k = \pi\,(k{-}1)/N_E$. Similarly, I neurons are arranged to cover the range uniformly, in an ordered manner corresponding to their IDs.

Nearby neurons on the ring will have smaller $\Delta\theta = \theta_i{-}\theta_j$, which translates to stronger weights than random, with neurons far away from each other by $\Delta\theta = \pi/4$ remaining at the nonspecific levels, and even farther pairs reducing their weights compared to random, with neurons with $\Delta\theta = \pi/2$ distance having the most negative modulation. Modulation of connection weights by parameter $m$, therefore, emulates the subnetwork structure in a continuous manner. Note that $m_{YX} = 0$ recovers the nonspecific weight condition described in Eq. (2). Unless stated otherwise, the parameters are chosen as: $\epsilon_{EE} = 0.01$, $\epsilon_{IE} = \epsilon_{EI} = 0.5$, $\epsilon_{II} = 0.85$, $J_{EE} = J_{IE} = 0.002$, $J_{EI} = J_{II} = -0.02$, $m_{EE} = m_{EI} = m_{IE} = 1$, $m_{II} = 0$.

To simulate the effect of single interneuron perturbations in our model, we perturb inhibitory neurons individually and measure its impact on the rest of neurons. The external input to individual I neuron is increased, from its baseline level, by $\delta s$.

The activity of the network (Eq. (1)) is simulated for $T_{sim}$, with and without this individual perturbation. The average activity of non-perturbed neurons is calculated, after discarding transient responses ($T_{trans}$). The change in the average activity of each neuron between the two conditions is obtained, $\delta r$, and the influence is quantified as the normalized changes of activity following perturbations: $\delta r/\delta s$. This procedure is repeated for all $N_I$ inhibitory neurons, and the distributions of influences are obtained for non-perturbed E and I neurons. The fraction of neurons in each subpopulation showing positive ($\delta r{>}0$) or negative ($\delta r{<}0$) changes are then quantified. Unless stated otherwise, default parameters are: $\delta s = 1$, $T_{sim} = 150$, $T_{trans} = 50$.

To measure the effect of external stimulation on our model networks, we stimulated a fraction (20%) of E or I neurons, which were chosen to be proximal on the ring (belonging to similar subnetworks). Stimulations were delivered as synchronous increases of the input to the selected neurons by $\delta s$ within a total stimulation time of $T_{stim}$. Stimulus was turned on for $\Delta T_{ON}$ (the input to stimulated neurons were increased to $s_b{+}\delta s$) and turned off for $\Delta T_{OFF}$ (the input to all neurons went back to the baseline level, $s_b$) in between. The average changes in the activity of other, nonstimulated neurons were obtained by calculating the mean response during stimulation, after discarding the transient responses (the initial $T_{trans}$). This was compared to the average responses, which were obtained from independent simulations without stimulation. The difference of the responses was obtained as changes in the activity following stimulation for each neuron (as shown in Figs 3D and S4b). We also calculated the pairwise correlation of activity between all pairs of neurons during the stimulation (which is shown in Figs 3D and S4b).

The parameters are chosen as: $\delta s = 1$, $T_{stim} = 1000$, $\Delta T_{ON} = 50$, $\Delta T_{OFF} = 150$, $T_{trans} = 50$, $\epsilon_{EE} = 0.01$, $\epsilon_{IE} = \epsilon_{EI} = 0.5$, $\epsilon_{II} = 0.85$, $m_{EE} = m_{EI} = m_{IE} = 1$, $m_{II} = 0$, $J_{EE} = J_{IE} = 0.002$, $J_{EI} = J_{II} = -0.01$.

The responses of the rate-based network model (Eq. (1)) in the stationary state can be analyzed by letting $dr_E/dt = 0$ and $dr_I/dt = 0$, leading to:

$$r_E = \Phi[W_{EE}r_E + W_{EI}r_I + s_E] \tag{4}$$

$$r_I = \Phi[W_{IE}r_E + W_{II}r_I + s_I]. $$

Changes in the stationary state responses upon external perturbations ($s+\delta s$) can be obtained from linearized dynamics of the network about the equilibrium point:

$$\delta r = (I - W)^{-1}\delta s, \tag{5}$$

where $\delta s$ is the vector of all input changes, $\delta r$ is the vector of all response changes, and $W$ is the total weight matrix of the network, describing all the connections between E and I neurons. The effect of single inhibitory neuron perturbations can be evaluated from Eq. (5), when $\delta s$ consists of 0s for all entries except for the perturbed neuron. We numerically solve Eq. (5) for each inhibitory neuron perturbation and repeat the procedure for all inhibitory neurons to obtain similar measures of response changes as rate-based simulations. Our results from this linear analysis were in good match with the results obtained from the perturbations of rate-based dynamics (S4a Fig), suggesting our findings can be understood in terms of the structure of weight matrices.

## Statistics

Statistical tests were performed in Python or MATLAB (R2022b). Pairwise comparisons between distributions were performed using the Mann–Whitney U test for unpaired groups and with the Wilcoxon signed rank test or $t$ tests for paired groups. For pairwise correlations between neurons' activities, we used Pearson's correlations. To compare the proportion of positively and negatively modulated cells in the all-optical experiments, we used the Barnard's test.

## Supporting information

**S1 Fig. Validation of the GAD67-Cre transgenic mouse line to image CA1 interneurons (related to Figs 1, 2, and 4). (a)** Td-Tomato (Ai14) neurons are distributed in all the CA1 layers and show the same pattern as GAD67-expressing cells. (**b, c**) Ai14 cells are immunopositive (arrows) for GAD67 and GABA as shown in stratum pyramidale (sp) and stratum oriens (so). (**d–f**) Some parvalbumin (PV), neuronal nitric oxide synthase (nNos), or calretinin (CR) cells are also expressing Ai14 (arrows in d–f, respectively). (**g, h**) Injection of cre-dependent ST-ChroME virus induces labeling of GABA neurons distributed as expected in all the layers of CA1, injection of GCaMP6m induces green fluorescent protein expression in both pyramidal neurons and some ST-ChroME positive cells (arrows). (**i–l**) GAD67 immunolabeling confirms that ST-ChroME-positive cells are GABAergic neurons (arrows). slm, stratum lacunosum-moleculare; sr, stratum radiatum, ml, stratum moleculare. Scale bars, a, g: 100 μm; b, e, f, h–l: 20 μm.
(PDF)

**S2 Fig. Activity of pyramidal cells and interneurons in relation to locomotion and SCEs (related to Fig 1). (a)** Calcium traces from approximately 7 min of recording from all interneurons in a representative field of view (10 interneurons in total). (**b**) Same as Fig 1C, but restricted to rest or locomotion periods. Interneurons show higher activity than pyramidal cells during rest periods (b1, $p = 0.009$), but not locomotion periods (b2, $p = 0.115$, both

Wilcoxon signed rank tests, $n = 11$ FOVs from 6 mice). (**c**) Pairwise correlations between interneurons are significantly higher than the ones between pyramidal neurons ($p = 0.041$, Wilcoxon signed rank test, $n = 11$ FOVs from 6 mice) even when subsampling pyramidal cells to match interneurons' ΔF/Fs (to control for the higher ΔF/F of interneurons). (**d**) Linear model fitted for pyramidal-pyramidal vs. pyramidal-interneurons pairwise correlations (see Fig 1H) for individual recordings: proportion of recordings displaying significant fit ($p < 0.05$). (**e**) Distribution of the proportion of SCEs to which each cell participates. Both pyramidal cell (d1) and interneuron (d2) distribution show lognormal shapes. Lognormal fits are depicted in gray. (**f1**) Pyramidal cells that are highly active in SCEs (scoring above the 90th percentile in the distribution of SCE participation including all pyramidal cells) display significantly higher ΔF/F than other pyramidal cells ($p = 0.008$, Mann–Whitney U test, $n = 276$ highly active cells, $n = 2,517$ other cells, from 11 FOVs from 6 mice). (**f2**) Pyramidal cells that are highly active in SCEs have significantly higher pairwise Pearson's correlations to interneurons compared to other cells even when other cells are subsampled to match highly active cells' ΔF/Fs ($p = 9.7e^{-14}$, Mann–Whitney U test, $n = 276$ highly active cells, $n = 275$ other cells, from 11 FOVs from 6 mice). (**g1**) Pyramidal cells that are part of cell assemblies (IN) display significantly higher ΔF/F than pyramidal cells not in assemblies (OUT, $p = 4.2e^{-6}$, Mann–Whitney U test, $n = 361$ in assemblies, $n = 1,437$ not in assemblies, from 11 FOVs from 6 mice). (**g2**) Pyramidal cells that are part of cell assemblies (IN) display significantly higher ΔF/F than pyramidal cells not in assemblies (OUT, $p = 5.6e^{-6}$, Mann–Whitney U test, $n = 361$ in assemblies, $n = 319$ not in assemblies, from 11 FOVs from 6 mice). $^{*} p < 0.05$; $^{**} p < 0.01$. Boxplots represent medians (center) and interquartile ranges (bounds). The whiskers extend to the most extreme data points not considered outliers, which are plotted individually using the circles. Underlying data can be found in S5 Data.
(TIF)

**S3 Fig. Indirectly modulated cells: spatial distribution, correlation, and link to success rate of the stimulated neuron (related to Fig 2).** (**a**) Distribution of the distance between the stimulated interneuron and negatively (red), positively (blue), or unmodulated (gray) cells (Kruskal–Wallis H-test, 3 groups, $p = 0.19$). (**b**) Box plot indicates the correlation between the fluorescence calcium traces of positively (blue), negatively (red), and unmodulated (gray) neurons (Kruskal–Wallis H-test, 3 groups, $p = 0.17$). (**c**) Scatter plot with linear regression best-fit line indicating a significant correlation between the stimulation success rate and the mean baseline dF/F signal of the stimulated neuron (Pearson r = 0.206, $p = 0.012$). (**d**) Scatter plot with linear regression best-fit line indicating a correlation between the fraction of positively (**d1**) or negatively (**d2**) modulated neurons and the success rate of the target cell (Pearson's r = 0.251, $p = 0.002$; Pearson's r = 0.295, $p = 0.0003$, respectively). Boxplots represent median (center) and interquartile ranges (bounds). The whiskers extend to the most extreme data points not considered outliers, which are plotted individually using the circles. Underlying data can be found in S6 Data.
(TIF)

**S4 Fig. Further characterization of connectivity in the network model (related to Fig 3).** (**a**) Fractions of E and I neurons showing a net positive or negative change in their activity, as a result of single I perturbations. (**a1**) Results obtained when the effect is assessed from a linear analysis of the network dynamics and its weight matrix (W). (**a2**) Results obtained when I-I connections display the same specificity as E-I connections (m_II = 1). (**b**) Same as Fig 3D, but with external stimulation of inhibitory neurons (20 I neurons in the middle; the changes in the activity of stimulated I neurons are not shown). Underlying data can be found in S7 Data.
(TIF)

**S5 Fig. Robustness of the modeling results to network parameters (related to Fig 3).** The results of our single inhibitory neuron perturbations were robust to the choice of network parameters, and fine-tuning was not needed to obtain the key results. To show the robustness of our results to the change of parameters, we simulated our networks with different ranges of parameters and calculated the modulations in each network. (**b**) We decreased the main coupling in the network (J) by half or increased it twice and observed similar results. (**c**) We also changed the connection probability of E-E and E-I connections. We increased the connection probability of initially sparse E-E connections, from 1% to 10%, and observed similar results. (**d**) We also decreased the connection probability of E-I connections, from the original 50% to 25%, and observed similar results. We therefore conclude that our results are robust to the choice of parameters in the network as our results hold for a wide range of parameter space. Underlying data can be found in S8 Data.
(JPG)

**S6 Fig. Relationship between interneuron activity and cell assemblies (related to Fig 4).** (**a**) Interneurons are not clustered into single assemblies, as evidenced by the fact that pairwise correlations between interneurons are not higher than correlations between interneurons and pyramidal cells ($p = 0.16791$, Wilcoxon signed rank tests, $n = 11$ FOVs from 6 mice). (**b**) Contour maps indicating the centroids of all active neurons in a representative imaging session with 4 cell assemblies (SCE-based method). Pyramidal cells are depicted in blue, interneurons in red. Filled contours belong to an individual assembly, whereas empty contours do not. Note the lack of spatial clustering: pyramidal cells and interneurons, as well as cells forming and not forming assemblies, are intermingled. (**c1**) Assembly patterns (from 3 significant cell assemblies) were obtained from a representative recording (same imaging session as c1) using the PCA/ICA assembly detection method (see Methods for details). Each plot represents a significant principal component (assembly), with a given weight for each neuron. Interneurons' weights are depicted in red, pyramidal cells' weights in blue. (**c2**) Pyramidal cells and interneurons display similar assembly weights ($p = 0.4$, Wilcoxon signed rank test, $n = 31$ assemblies from 11 recordings from 6 mice). (**c3**) Fit of linear model between pyramidal cell assembly weight and interneuron assembly weight for each recording (averaged across 31 assemblies, $n = 11$ recordings, 6 mice). (**c4**) Pyramidal cell assembly weights are more clustered in a single assembly compared to interneurons (maximum assembly weight across assemblies divided by the absolute average of assembly weights; $p = 0.0025$, Mann–Whitney U test, $n = 31$ assemblies, from 11 FOVs from 6 mice). (**d1**) Lack of evidence of cell assembly segregation by single interneurons. Assembly activation-triggered average of pyramidal cells' calcium traces when each interneuron in an assembly is active (left) or inactive (right). Shaded areas represent standard deviations. Top, traces from pyramidal cells in the same assembly as the interneuron. Middle, traces from pyramidal cells in different assemblies. Bottom, traces from pyramidal cells not forming assemblies. Note that the activity of the interneuron in an assembly does not affect the activity of pyramidal cells of competing assemblies or of the ones not forming assemblies. (**d2**) No significant difference in ΔF/F peak at assembly activation for pyramidal cells when the interneuron in the same assembly is active or inactive ($p = 0.7$, Wilcoxon signed-rank test, $n = 7$ recordings with significant assemblies from 5 mice). Data from 21 cell assemblies. ** $p < 0.01$. Boxplots represent medians (center) and interquartile ranges (bounds). The whiskers extend to the most extreme data points not considered outliers, which are plotted individually using the circles. Underlying data can be found in S9 Data.
(TIF)

**S7 Fig. Changes in SCE frequency and amplitude are absent if targeted interneuron does not respond to light stimulation (related to Fig 4).** (**a1**) Same as Fig 4B_3, but for all the

experiments, excluding the experiments with unresponsive cells, $p = 0.035$. (**a2**) Same as Fig 4B$_4$, but for all the experiments, excluding the experiments with unresponsive cells, $p = 0.0007$. (**b1**) Same as Fig 4B$_3$, but for the experiments with unresponsive cells only, $p = 0.45$. (**b2**) Same as Fig 4B$_4$, but for the experiments with unresponsive cells only, $p = 0.13$ (Mann–Whitney U test in all cases). * $p < 0.05$; *** $p < 0.001$. Boxplots represent medians (center) and interquartile ranges (bounds). The whiskers extend to the most extreme data points not considered outliers, which are plotted individually using the circles. Underlying data can be found in S10 data.
(TIF)

**S1 Table. Table of the distribution of all-optical experiments and modulated cells per animal.**
(DOCX)

**S1 Movie. Effect of single interneuron photoactivation on SCEs.** Representative movie of a stimulation experiment corresponding to Fig 4B. Synchronous calcium events (SCEs) were recorded in the CA1 region of a GAD67-Cre adult mouse, injected with AAV1.Syn-GCaMP6m and AAV9.CAG.DIO.ChroME-ST.mRuby3 at birth. The movie (top) was recorded at 8.4 Hz in a field of view of $280 \times 280$ μm$^2$. The baseline period was followed by a stimulation period and a poststimulation period. The stimulated cell corresponds to the cell illustrated in Fig 2C. At the frame corresponding to stimulation, the cell is highlighted with a white rectangle. Presented alongside the movie above is a raster plot that displays the average calcium fluorescence signal from all imaged neurons over time. This raster plot shows vertical white stripes that identify SCEs. They occur at a higher rate during the stimulation period.
(MOV)

**S1 Data. Supporting data for Fig 1.**
(XLSX)

**S2 Data. Supporting data for Fig 2.**
(XLSX)

**S3 Data. Supporting data for Fig 3.**
(XLSX)

**S4 Data. Supporting data for Fig 4.**
(XLSX)

**S5 Data. Supporting data for S2 Fig.**
(XLSX)

**S6 Data. Supporting data for S3 Fig.**
(XLSX)

**S7 Data. Supporting data for S4 Fig.**
(XLSX)

**S8 Data. Supporting data for S5 Fig.**
(XLSX)

**S9 Data. Supporting data for S6 Fig.**
(XLSX)

**S10 Data. Supporting data for S7 Fig.**
(XLSX)

## Acknowledgments

We thank all the members of the Cossart lab for helpful discussions and constructive feedback. We thank INMED's animal facility and PBMC technological platform for excellent technical support.

## Author Contributions

**Conceptualization:** Marco Bocchio, Rosa Cossart.

**Data curation:** Marco Bocchio, Artem Vorobyev, Susanne Reichinnek, Agnes Baude, Rosa Cossart.

**Formal analysis:** Artem Vorobyev, Sadra Sadeh, Robin Dard, Agnes Baude, Claudia Clopath, Rosa Cossart.

**Funding acquisition:** Rosa Cossart.

**Investigation:** Marco Bocchio, Artem Vorobyev, Sadra Sadeh, Claudia Clopath, Rosa Cossart.

**Methodology:** Marco Bocchio, Artem Vorobyev, Sadra Sadeh, Sophie Brustlein, Robin Dard, Valentina Emiliani, Claudia Clopath, Rosa Cossart.

**Project administration:** Rosa Cossart.

**Resources:** Claudia Clopath, Rosa Cossart.

**Supervision:** Rosa Cossart.

**Validation:** Rosa Cossart.

**Visualization:** Rosa Cossart.

**Writing – original draft:** Marco Bocchio, Artem Vorobyev, Sadra Sadeh, Claudia Clopath, Rosa Cossart.

**Writing – review & editing:** Marco Bocchio, Artem Vorobyev, Sadra Sadeh, Agnes Baude, Claudia Clopath, Rosa Cossart.

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
