## [Editor Report · Decision Letter 0]

5 Jun 2024

Dear Rosa, 

Thank you for submitting your manuscript entitled "Functional networks of inhibitory neurons orchestrate synchrony in the hippocampus", which has been revised in response to reviews from Nature Communications, for consideration as a Research Article by PLOS Biology.

We have now received the review history from Nature Communications, and we have had a chance to discuss your revision with an Academic Editor with relevant expertise. I am writing to let you know that we are, in principle, interested in the study and we are largely satisfied by your responses to Reviewer 1. However, we would like to send your submission back to reviewers 2 and 3 for their feedback on the revision, to make sure they are satisfied by the changes made.

Once your full submission is complete, your paper will undergo a series of checks in preparation for peer review. After your manuscript has passed the checks it will be sent out for review. To provide the metadata for your submission, please Login to Editorial Manager (https://www.editorialmanager.com/pbiology) within two working days, i.e. by Jun 07 2024 11:59PM.

Kind regards,

Luke

Lucas Smith, Ph.D.

Senior Editor

PLOS Biology

lsmith@plos.org

---

## [Decision Letter · Decision Letter 1]

10 Jul 2024

Dear Rosa,

Thank you for your patience while we considered your revised manuscript "Functional networks of inhibitory neurons orchestrate synchrony in the hippocampus" for consideration as a Research Article at PLOS Biology. Your revised study has now been evaluated by the PLOS Biology editors, the Academic Editor and one of the original reviewers. 

In light of the reviews, which you will find at the end of this email, we are pleased to offer you the opportunity to address the remaining points from Reviewer 1 (their own comment and their comments 2, 8, 11 on Reviewer 2's comments) in a revision that we anticipate should not take you very long. We will then assess your revised manuscript and your response to the reviewers' comments with our Academic Editor aiming to avoid further rounds of peer-review, although might need to consult with the reviewers, depending on the nature of the revisions.

In addition, we have a few editorial requests which we would like you to address:

* Please add the links to the funding agencies in the Financial Disclosure statement in the manuscript details

* DATA POLICY:

Regardless of the method selected, please ensure that you provide the individual numerical values that underlie the summary data displayed in the following figure panels as they are essential for readers to assess your analysis and to reproduce it: 1CDEGHI2I4, 2D, 3ABC, 4AB, S2BCE, S3AB, S4A, S5, S6, S7ACD and S8AB

* CODE POLICY

* Please note that per journal policy, we do not allow the mention of "data not shown", "personal communication", "manuscript in preparation" or other references to data that is not publicly available or contained within this manuscript. Please either remove mention of these data or provide figures presenting the results and the data underlying the figure(s).

* Please note that per journal policy, the model system/species studied should be clearly stated in the abstract of your manuscript. 

**IMPORTANT - SUBMITTING YOUR REVISION**

*Resubmission Checklist*

*Published Peer Review*

*PLOS Data Policy*

*Blot and Gel Data Policy*

Sincerely,

Christian

Christian Schnell, PhD

Senior Editor

PLOS Biology

cschnell@plos.org

REVIEWS:

Reviewer #1: In general, I believe that the manuscript has improved and that most of my comments have been addressed properly. A last minor point:

- Low and high firing rate neurons can indeed bias pairwise correlations or other rate-derived metrics in both extracellular recordings and imaging data. Consequently, a decrease in statistical power, as found by the authors, is expected. However, I believe this is a relevant result, highlighting the specificity for interneurons in the analysis shown in Fig. 1c. I still think a similar analysis would be a desirable addition to some important panels for the narrative, such as Fig. 1i.

I believe that most of the comments from reviewer 3 have been either correctly addressed or thoroughly discussed.

- Point 1. The new sections are largely different from the previous manuscripts and are now more clear and comprehensive.

- Point 2. The point raised by Reviewer 3 is both valid and interesting, reflecting an ongoing debate in the field. It is well-established that CA1 neurons largely inherit their firing patterns from upstream CA3 and EC networks (Ahmed and Mehta, 2009; Brun et al., 2008; Franzius et al., 2007; Rolls et al., 2006; Savelli and Knierim, 2010; Solstad et al., 2006; Steffenach et al., 2005). However, recent experiments have shown that a significant fraction of CA1 pyramidal neurons remain place cells even in the absence of major EC and CA3 inputs (Zutshi et al, 2022 Neuron; in fact, this paper is specifically exploring this question: "Extrinsic control and intrinsic computation in the hippocampal CA1 circuit" ), and this results align with previous and current observations (Huszar et al., 2022, Nat Neurosci; Stark et al., 2013, Neuron). These findings suggest that the CA1 circuitry is intrinsically shaped by ontogeny (Farooq and Dragoi, 2019, Science; Huszar et al., 2022, Nat Neurosci) and experience (Wilson and McNaughton, 1994, Science), enabling it to parse and describe the variance of inputs received from CA3 and EC as part of its spontaneous activity. The work of Bocchio et al. further investigates the inhibitory-excitatory synapses in CA1, which are major contributors to these intrinsic computations.

- Point 3: Terada et al. 2022 is cited, and SWR references are toned down.

- Point 4. I concur with the reviewer comment about the fact that focusing on interneurons residing in the pyramidal layer restricts the experiment to mostly perisomatic targeting of interneurons. The two neurons increasing activity when locomotion stops can well be CCK-basket cells (Dudok et al., 2021 Neuron). I believe that the authors addressed this point correctly.

- Point 5. Based on the overall results of the work, the new paragraph about interneuron types underlying local disinhibition is an important addition.

- Point 6. In my opinion, and in light of the author's comment, the inclusion of gap-junction-mediated synchrony, which is potentially interesting, is out of the scope of this manuscript.

- Point 7. The supplementary table certainly improves data interpretability and transparency and, in my opinion, addresses the reviewer concern.

- Point 8. I believe that including some of the statistics of the network model (Fig. S5) as a main panel in figure 3 will strengthen the message of this figure.

- Point 9. Gad67 selectivity for GABergic types in CA1 is a major concern of reviewer 1 and has been extensively addressed by the authors.

- Point10. Correctly addressed by the authors, in my view.

- Point 11. In the current version, the authors have included in the text a statistical statement for Figure 4c: "All of the pairs recruited in the same assembly during or just after stimulation were already part of a similar assembly during baseline (median 100%, n = 236 pairs, 3 mice)," but I feel that a more comprehensive visual description of this important result (including statistics) will improve the manuscript.

---

## [Editor Report · Decision Letter 2]

6 Sep 2024

Dear Dr Cossart,

Thank you for the submission of your revised Research Article "Functional networks of inhibitory neurons orchestrate synchrony in the hippocampus" for publication in PLOS Biology. On behalf of my colleagues and the Academic Editor, Jozsef Csicsvari, I am pleased to say that we can in principle accept your manuscript for publication, provided you address any remaining formatting and reporting issues. These will be detailed in an email you should receive within 2-3 business days from our colleagues in the journal operations team; no action is required from you until then. Please note that we will not be able to formally accept your manuscript and schedule it for publication until you have completed any requested changes.

PRESS

Sincerely, 

Suzanne

Suzanne De Bruijn, PhD, 

Associate Editor

PLOS Biology

sbruijn@plos.org